# Is Your Paper Being Reviewed by an LLM? Benchmarking AI Text Detection in Peer Review

**Sungduk Yu**
Oracle AI
sungduk.yu@oracle.com

**Man Luo**
Abridge
man.luo@abridge.com

**Avinash Madusu**
Intel Labs
avinash.madusu@intel.com

**Vasudev Lal**
Oracle AI
vasudev.lal@oracle.com

**Phillip Howard**
Thoughtworks
phillip.howard@thoughtworks.com

## Abstract

Peer review is a critical process for ensuring the integrity of published scientific research. Confidence in this process is predicated on the assumption that experts in the relevant domain give careful consideration to the merits of manuscripts which are submitted for publication. With the recent rapid advancements in large language models (LLMs), a new risk to the peer review process is that negligent reviewers will rely on LLMs to perform the often time consuming process of reviewing a paper. However, there is a lack of existing resources for benchmarking the detectability of AI text in the domain of peer review. To address this deficiency, we introduce a comprehensive dataset containing a total of 788,984 AI-written peer reviews paired with corresponding human reviews, covering 8 years of papers submitted to each of two leading AI research conferences (ICLR and NeurIPS). We use this new resource to evaluate the ability of 18 existing AI text detection algorithms to distinguish between peer reviews fully written by humans and different state-of-the-art LLMs. Additionally, we explore a context-aware detection method called Anchor, which leverages manuscript content to detect AI-generated reviews, and analyze the sensitivity of detection models to LLM-assisted editing of human-written text. Our work reveals the difficulty of identifying AI-generated text at the individual peer review level, highlighting the urgent need for new tools and methods to detect this unethical use of generative AI. Our dataset is publicly available at: `https://huggingface.co/datasets/IntelLabs/AI-Peer-Review-Detection-Benchmark`.

## 1 Introduction

Recent advancements in large language models (LLMs) have enabled their application to a broad range of domains, where LLMs have demonstrated the ability to produce plausible and authoritative responses to queries even in highly technical subject areas. These advancements have coincided with a surge in interest in AI research, resulting in increased paper submissions to leading AI conferences (Audibert et al., 2022). Consequently, workloads for peer reviewers have also increased significantly, which could make LLMs an appealing tool for lessening the burden of fulfilling their peer review obligations (Kuznetsov et al., 2024; Kousha & Thelwall, 2024; Zhuang et al., 2025).

Despite their impressive capabilities, the use of LLMs in the peer review process raises several ethical and methodological concerns which could compromise the integrity of the publication process (Hosseini & Horbach, 2023; Latona et al., 2024; Seghier, 2024; Zhou et al., 2024). Reviewers are selected based on their expertise in a technical domain related to a submitted manuscript, which is necessary to critically evaluate the proposed research. Offloading this responsibility to an LLM circumvents the role that reviewer selection plays in ensuring proper vetting of a manuscript. Furthermore, LLMs are prone to hallucination and may not possess the ability to rigorously evaluate research publications. Therefore, the use of LLMs in an undisclosed manner in peer review poses a significant ethical concern that could undermine confidence in this important process.

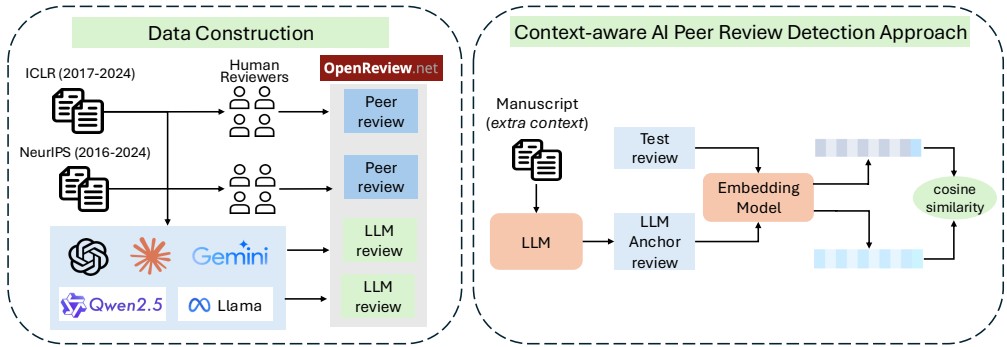

Figure 1: Left panel: our data construction pipeline. Right panel: our context-aware detection method (Anchor) specifically designed for AI-generated review detection evaluated in Section 4.1.

Motivating the need for evaluation resources and detection tools to address this problem is the apparent increase in AI-generated text among peer reviews submitted to recent AI research conferences. Prior studies revealed an upward trend in AI-generated texts among peer reviews (Liang et al., 2024a; Latona et al., 2024). This trend is particularly concerning given that evaluations from human and AI reviewers are not aligned (Drori & Te'eni, 2024; Latona et al., 2024; Ye et al., 2024) and that LLM-generated reviews lack robustness (Li et al., 2025), suggesting the unregulated and undisclosed use of LLMs in peer review could undermine the integrity of the current system.

Despite the growing recognition of this problem, there is a lack of existing dataset resources for comprehensively evaluating the performance of AI text detection methods in the domain of peer review. To address this deficiency, we introduce the largest dataset to-date of parallel human-written and LLM-written peer reviews for 8 years of papers submitted to two leading AI research conferences, NeurIPS and ICLR (Figure 1). Our dataset consolidates human-written peer reviews from existing sources with AI-written peer reviews that we generated for the same paper using five state-of-the-art LLMs: GPT-4o Achiam et al. (2023), Claude Sonnet 3.5 Anthropic (2023), Gemini 1.5 pro Team et al. (2023), Qwen 2.5 72b Bai et al. (2023), and Llama 3.1 70b Dubey et al. (2024). In total, our dataset contains 788,984 peer reviews, evenly balanced between human-written reviews and AI-generated peer reviews created by these five LLMs.

We use our dataset to investigate the suitability of various AI text detection methods for identifying LLM generations in the peer review process. While limited prior work has analyzed the presence of AI-generated text in peer reviews at the corpus level (Liang et al., 2024a) or has analyzed the use of propriety solutions (Latona et al., 2024), our study is the first to investigate the detectability LLM generations at the individual review level using synthetically generated AI samples, which is necessary to address this problem in practice. Specifically, we evaluate 18 existing open-source methods for AI text detection.

Our results show that most existing AI-text detection methods are limited in their ability to robustly detect AI-generated reviews while maintaining a low number of false positives. Motivated by this finding, we test an alternative approach which is specifically designed for AI text detection in the peer review context. Leveraging the additional context available in the peer review setting, our method detects AI-generated peer reviews by comparing the semantic similarity of a given review to a set of reference AI-generated reviews for the same paper. We find that this simple yet effective method surpasses the performance of all existing approaches in detecting GPT-4o and Claude written peer reviews. Additionally, we conduct analyses to understand how different levels of AI use for editing reviews impacts detectability and false positives, as well as the characteristics which distinguish LLM-written peer reviews from those written by humans. Our work demonstrate the challenge of detecting AI-written text in peer reviews and motivates the need for further research on methods to address this unethical use of LLMs in the peer review process.

To summarize, our contributions are as follows: (1) We publicly release a dataset of 788,984 AI-written peer reviews generated by five widely-used LLMs paired with human-written reviews for the same papers, which is the largest resource to-date for studying AI text detection in peer review. (2) Using our dataset, we benchmark 18 open-source AI text detection algorithms, finding that most struggle to reliably detect fully AI-written peer reviews at low false positive rates. (3) We propose a

new *context-aware* detection method which compares the semantic similarity between a candidate review and a reference LLM-generated review for the same paper, achieving strong performance under strict FPR constraints. (4) We conduct analyses revealing key differences between human- and AI-written reviews, finding that AI-generated reviews are generally less specific, more favorable, and more confident. (5) We evaluate how LLM-assisted editing in peer review affects detection rates.

## 2 RELATED WORK

**AI text detection datasets**   Several datasets have been introduced to evaluate AI text detection models. RAID-TD  (Dugan et al., 2024) provides a large-scale benchmark designed to assess text detection under adversarial conditions, ensuring robustness against manipulated AI-generated content. The M4 Dataset  (Wang et al., 2024) expands the scope by incorporating reviews from multiple LLMs across different languages, offering a more diverse linguistic evaluation. The HC3 Dataset  (Guo et al., 2023) consists of responses from ChatGPT and human experts, covering specialized domains such as finance, medicine, and law, in addition to general open-domain content. In contrast, the GPT Reddit Dataset (GRiD)  (Qazi et al., 2024) focuses on social media conversations, compiling a diverse set of human- and AI-generated responses to Reddit discussions. Meanwhile, Beemo (Artemova et al., 2024) introduces a benchmark of expert-edited machine-generated outputs, spanning creative writing, summarization, and other practical applications. These benchmarks primarily evaluate AI-generated text from a single model and do not address the domain of AI text in peer review. In contrast, our dataset is larger than most existing datasets (788k generations) and is unique in its focus on AI text detection in peer review.

**AI-generated text detection**   AI-generated text detection has been framed as a binary classification task to distinguish human-written from machine-generated text  (Bakhtin et al., 2019; Jawahar et al., 2020; Fagni et al., 2021; Mitchell et al., 2023b). Solaiman et al. (2019) used a bag-of-words model with logistic regression for GPT-2 detection, while fine-tuned language models like RoBERTa (Liu et al., 1907) improved accuracy  (Zellers et al., 2019; Uchendu et al., 2020; Gehrmann et al., 2019). Zero-shot methods based on perplexity and entropy emerged as alternatives  (Ippolito et al., 2020; Gehrmann et al., 2019). Other studies focused on linguistic patterns and syntactic features for model-agnostic detection  (Uchendu et al., 2020; Gehrmann et al., 2019). Watermarking techniques, such as DetectGPT  (Mitchell et al., 2023b), have also been proposed for proactive identification. Centralized frameworks like MGTBench  (He et al., 2023) and its refined version, IMGTB  (Spiegel & Macko, 2023), provide standardized evaluations for AI text detection. IMGTB categorizes methods into model-based and metric-based approaches. Model-based methods leverage large language models such as ChatGPT-turbo  (OpenAI, 2023) and Claude  (Anthropic, 2023). Metric-based methods, including Log-Likelihood  (Solaiman et al., 2019), Rank  (Gehrmann et al., 2019), Entropy (Gehrmann et al., 2019), DetectGPT  (Mitchell et al., 2023b), and DetectLLM  (Su et al.), rely on log-likelihood and ranking for classification.

While there is some similarity between our Anchor approach and methods such as DetectLLM and DNA-GPT (Yang et al., 2023), there are key differences in both the functionality and applicability of our method relative to this prior work. DetectLLM assumes a white-box detection setting where the detector has access to the LLM used to generate the evaluated text. Unfortunately this setting is unrealistic for our task of detecting AI text in peer review because there is no way of knowing which LLM was used to generate the review. Additionally, we view frontier commercial models as the most likely to be used in practice due to their superior capabilities; since these models are only available via an API, methods such as DetectLLM which require full access to the model are not compatible.

Another key difference between our Anchor method and both DetectLLM and DNA-GPT is that it leverages additional context which is available in the peer review setting: the manuscript of the paper which is being reviewed. In contrast, DNA-GPT truncates the middle of an evaluated text, regenerates the remainder with an LLM, and then analyzes the difference between the original & regenerated portions via n-gram analysis (in the black-box detection setting). This is unlikely to produce text which is as representative of AI-generated reviews because it lacks grounding in the source content (the paper). To the best of our knowledge, our anchor embedding approach is unique among AI text detection methods in its use of the additional context available in the peer review setting.

**AI-assisted peer review**   Recent studies have explored the role of LLMs in peer review, examining their influence on reviewing practices (Liang et al., 2024a;b), simulating multi-turn interactions (Tan et al., 2024), and assessing their reviewing capabilities (Zhou et al., 2024). Tools like OpenReviewer (Tyser et al.) provide AI-assisted review improvements, while other works focus on LLM transparency (Kuznetsov et al., 2024) and distinguishing AI-generated content (Mosca et al., 2023). Recent studies have investigated AI-driven review systems (Tyser et al., 2024), agentic frameworks (D'Arcy et al., 2024), and comparative analyses of LLM-generated reviews (Liu & Shah, 2023), along with broader explorations of LLMs' roles and limitations in peer review (Jin et al., 2024; Santu et al., 2024; Sukpanichnant et al., 2024; Liang et al., 2024b). While it is not the primary focus of our work, we analyze the quality of LLM-generated peer reviews in Section 5.

## 3   DATASET CONSTRUCTION

### 3.1   HUMAN REVIEWS

We used the OpenReview API (OpenReview) to collect submitted manuscripts and their reviews for the ICLR conferences from 2019 to 2024, as well as for NeurIPS conferences from 2021 to 2024[1]. Additionally, we used the ASAP dataset Yuan et al. (2022) to collect manuscripts and reviews for ICLR 2017 to 2018 and NeurIPS 2016 to 2019.

### 3.2   AI REVIEWS

We generated 788,984 AI-generated reviews using five widely-used LLMs: GPT-4o, Claude Sonnet 3.5, Gemini 1.5 pro, Qwen 2.5 72b, and Llama 3.1 70b.

**Prompts.** To control the content and structure of these AI-generated reviews, we included conference-specific reviewer guidelines and review templates in the prompts. Note that review templates have evolved significantly over time (Table S1), necessitating prompt adaptations for papers submitted in different years. We additionally aligned the paper decisions by prompting the LLMs with specific decisions derived from the corresponding human reviews (see Appendix E for complete prompt details). This step is important, as we found that AI review content and recommendations vary substantially depending on how input prompts are constructed. Thus, these measures represent our efforts to control the influence of text prompts. Importantly, as we show later in Section 5.1, despite using a consistent prompting strategy, our dataset remains robust to prompt variation and supports generalizable detection performance across different prompting styles.

Throughout the course of our study, we evaluated multiple prompting strategies ranging from prompts which produce fully AI-written reviews to ones which request varying levels of LLM editing of human-written peer reviews (Section 5.2). Our primary focus is on the former scenario of detecting fully AI-written peer review, which we believe to be the most pressing ethical problem. However, we also included the LLM-edited human reviews (with varying levels of editing) from our analysis in Section 5.2 in our released dataset.

**Computation.** We used Azure OpenAI Service, Amazon Web Services, and Google Cloud Platform to generate GPT-4o, Claude, and Gemini reviews, respectively. For Qwen reviews, we used NVIDIA RTX 6000 GPUs, and for Llama reviews, we used Intel Gaudi 2 accelerators.

### 3.3   DATASET STATISTICS

Table 1 provides complete statistics for our generated dataset (see Appendix B.4 for a breakdown by conference year and review-generating LLM). We withheld a randomly sampled subset of reviews to serve as a calibration set, which is used in our experiments to determine classification thresholds for each evaluated method. This calibration set contains 75,824 AI-generated and human-generated peer reviews, divided approximately evenly across all five LLMs. To construct the calibration set, we randomly selected 500 papers from ICLR (2021, 2022) and NeurIPS (2021, 2022) and generated AI reviews corresponding to the human reviews for each paper. Because our sampling was done at the paper level rather than the review level, the number of reviews per paper—and consequently per conference—varies slightly. To facilitate the evaluation of detection methods which are more

---

[1]NeurIPS 2020 reviews are not publicly available

Table 1: Dataset statistics. Each subset in the dataset is balanced, containing an equal number of human-written and AI-generated peer reviews. The calibration and test sets include reviews generated by five LLMs, whereas the extended set includes reviews generated by GPT-4o and Llama 3.1.

| | Calibration set | | Test set | | Extended set | |
|---|---|---|---|---|---|---|
| | ICLR | NeurIPS | ICLR | NeurIPS | ICLR | NeurIPS |
| GPT-4o | 7,710 | 7,484 | 27,342 | 30,212 | 121,278 | 91,994 |
| Gemini 1.5 Pro | 7,704 | 7,476 | 27,278 | 30,146 | - | - |
| Claude Sonnet 3.5 v2 | 7,672 | 7,484 | 27,340 | 30,208 | - | - |
| Llama 3.1 70b | 7,708 | 7,472 | 27,284 | 30,086 | 121,130 | 91,706 |
| Qwen 2.5 72b | 7,662 | 7,452 | 27,190 | 29,966 | - | - |
| Total | | 75,824 | | 287,052 | | 426,108 |
| Grand total | | | | | | **788,984** |

computationally expensive (e.g., methods which requires using LLMs as surrogates), we also withheld a separate test set consisting of human reviews and those generated by all five LLMs for 500 randomly sampled papers from each conference & year. This test split contains a total of 287,052 reviews and is used throughout our main experimental results. A further 426,108 reviews were generated from GPT-4o and Llama 3.1 70b, which we designate as an extended set of reviews (additional experimental results for the extended set are provided in Appendix C.4).

While we include reviews from post-ChatGPT[2] conferences (ICLR 2023–2024 and NeurIPS 2023–2024), we note that the human-labeled reviews from these years may include LLM-assisted texts. These reviews are retained to support broader research use cases, such as longitudinal analysis of linguistic trends. However, we do not use them in our main experiments, which focus on pre-2023 data where we have higher confidence in the human-authored labels.

We note slight variations in sample sizes among the five LLMs. These differences stem from LLM-specific factors such as context window limits, generation errors (e.g., malformed JSON or degenerate outputs), and input prompt safety filters.

## 4 EXPERIMENTAL RESULTS

### 4.1 IMPROVING DETECTION OF AI-GENERATED PEER REVIEWS USING MANUSCRIPT CONTEXT

Building on our earlier anchor-based detection approach Yu et al. (2024), peer review provides additional contextual information for this problem: the manuscript being reviewed. With access to metadata connecting reviews to their source manuscripts (as in platforms like OpenReview.net), we investigate whether leveraging the manuscript can improve detection of AI-generated peer reviews. Concurrent work has explored similar semantic signals for this task Kumar et al. (2024).

To test this idea, we introduce a method which utilizes the manuscript by comparing the semantic similarity between a test reviews (TR) and a synthetic "Anchor Review" (AR) generated for the same manuscript. The AR can be generated by any LLM. We use a simple, generic prompt (Appendix E.2) to generate the AR without prior knowledge of the user prompts (i.e., the AR prompt differs from those used to create reviews in the testing dataset). Once an AR is generated for a given paper (Eq.1) and a testing review (TR) is provided, we obtain their embeddings using a text embedding model (EM, Eqs.2 and 3). The semantic similarity between the embeddings of the AR and TR is then computed via cosine similarity (Eq.4). Finally, this similarity score is compared against a learned threshold ($\theta$): if the score exceeds the threshold, the review is classified as AI-generated (Eq. 5).

(Formalization of Anchor Method)

$$AR = LLM(paper, Prompt_{AR}) \qquad (1)$$

$$Emb_{AR} = EM(AR) \qquad (2)$$

$$Emb_{TR} = EM(TR) \qquad (3)$$

$$Score = Cosine\_similarity(Emb_{AR}, Emb_{TR}) \qquad (4)$$

$$Label = \begin{cases} 1 & \text{if Score} > \theta, \\ 0 & \text{otherwise.} \end{cases} \qquad (5)$$

In our study, we use OpenAI's embedding model (text-embedding-003-small). The threshold $\theta$ is learned from the calibration set. Specifically, for each review in the calibration data, we apply the

---
[2]ChatGPT was released on November 30, 2022.

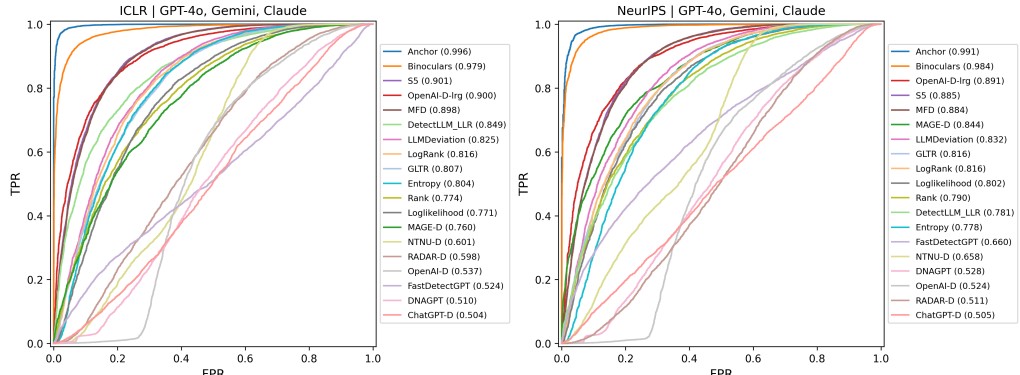

Figure 2: ROC plots computed from the combined GPT-4o, Gemini, and Claude review calibration dataset, showing results for ICLR (left) and NeurIPS (right); AUC values are shown in parentheses.

steps outlined in Eqs. 1 to 4. To handle cases where the source LLM is unknown, we generate multiple anchor reviews using different LLMs and apply a voting scheme: if any anchor yields a positive detection, the review is labeled as AI-generated (see Appendix D.1 for details).

## 4.2 FULLY AI-WRITTEN REVIEW DETECTABILITY

We compare our approach with 18 baseline methods using IMGTB (see Appendix A for details) and utilize the calibration set to determine appropriate thresholds for the test data. To minimize the risk of LLM-text contamination in human-written reviews, we only include reviews submitted before the release of ChatGPT in late 2022 (i.e., NeurIPS 2016-2022 and ICLR 2017-2022 papers), when LLM use rapidly became widespread. The threshold is then determined by setting a target False Positive Rate (FPR), which is achieved by adjusting the threshold until the FPR equals the target value. We focus on low FPR targets (e.g., 0.1%, 0.5%, and 1%) because false positive classifications—where human-written reviews are mistakenly identified as AI-generated—carry high stakes, potentially damaging an individual's reputation. Additionally, we focus on AI review text samples generated by three commercial LLMs (GPT-4o, Gemini, and Claude) because these models are more advanced, making the AI text detection task harder. Many users are more likely to choose these models over open-source LLMs due to their convenient user interfaces and limited access to the compute resources required for running advanced open-source models.

AI text detection models can be calibrated for varying levels of sensitivity in order to balance the trade-off between true positive and false positive detections. Receiver operating characteristic (ROC) curves are therefore commonly used to compare different methods, as they provide a visualization of the true-positive rate (TPR) which can be achieved by a model when its decision boundary is calibrated for a range of different FPRs. Figure 2 provides the ROC curves for baseline methods, calculated using our GPT-4o, Gemini, and Claude review calibration subset separately for reviews submitted to ICLR 2021-2022 (left) and NeurIPS 2021-2022 (right). The area under the curve (AUC) is provided for each method in the legend; higher values indicate better detection accuracy across the entire range of FPR values. Among the 18 baseline models, Binoculars consistently performs the best across the ROC curve, particularly excelling at maintaining a low FPR. Our task-specific anchor method appears to perform strongly, but we will discuss its performance separately in Section 4.1; here, we focus on analyzing the 18 existing detection methods.

Although ROC curves are useful for comparing the overall performance of different classifiers, only the far left portion of these plots are typically relevant for practical applications of AI text detection models. This is particularly true in the domain of peer review, where the cost of a false positive is high. Reviewers volunteer their time and expertise to sustain this important process; false accusations have the potential to further reduce the availability of reviewers due to disengagement and can also lead to significant reputational harm. Therefore, it is vital that AI text detection systems for peer review be calibrated for a low FPR in order to avoid such negative outcomes.

Prior work has shown that AUC is not necessarily reflective of how models perform at very low FPR values (Yang et al., 2023; Krishna et al., 2024; Tufts et al.). Therefore, we also re-

port the *actual* TPR and FPR achieved by different detection methods at discrete low values of target FPR (0.1%, 0.5%, and 1%), which we believe to be of greatest interest for practical applications. The target FPR is used to calculate each method's classification threshold using our calibration dataset, with the actual TPR and FPR computed over the withheld test dataset. To simulate a more challenging evaluation setting where some of the test reviews are "out-of-domain" in the sense that they come from a different conference than the calibration dataset, we use only the ICLR reviews to calibrate each method (see Section C.2 for in-domain evaluations).

Table 2 provides these results separately for the detection of GPT-4o, Gemini, and Claude reviews (Llama and Qwen results are provided in Appendix C.3). Other baseline methods that failed achieve a TPR of at least 1% at a target FPR of 1% (i.e., TPR < 1% at target FPR = 1%) are omitted.

The results in Table 2 show that among the 18 baselines, Binoculars generally achieves the best performance overall. For example, its TPR reaches 45–85% at a 1% target FPR, while other models reach only up to 19% TPR. However, even with the best-performing Binoculars, performance drops significantly at more stringent FPR levels—for instance, it achieves only 17% TPR at 0.1% FPR for GPT-4o reviews. These results highlight a key challenge in AI-generated peer review detection: the difficulty of robustly identifying AI-written text at the level of individual peer reviews. At the same time, they point to the need for new methods that can improve accuracy and robustness under low-FPR constraints. To this end, we also evaluate a purpose-built detection method (Anchor) as described previously in Section 4.1, which leverages additional context specific to the peer review detection task.

Compared to the 18 baseline models, our Anchor approach consistently achieves the highest

Table 2: Actual FPR and TPR calculated from the withheld test dataset at varying detection thresholds, which are calibrated using ICLR reviews from our calibration set at different target FPRs. Highest TPRs are in **bold**. The results are well separated; see Table S5 for uncertainties estimated via bootstrapping.

| Target FPR: | | 0.1% | | 0.5% | | 1% | |
|---|---|---|---|---|---|---|---|
| | | FPR | TPR | FPR | TPR | FPR | TPR |
| GPT-4o Reviews | Anchor | 0.1 | **63.5** | 0.5 | **83.7** | 1.0 | **88.8** |
| | Binoculars | 0.2 | 17.1 | 0.6 | 33.6 | 1.0 | 45.2 |
| | MAGE-D | 0.1 | 2.3 | 0.6 | 8.8 | 1.3 | 14.7 |
| | s5 | 0.1 | 0.1 | 0.9 | 7.2 | 1.7 | 17.5 |
| | MFD | 0.2 | 0.1 | 0.8 | 6.0 | 1.6 | 15.6 |
| | GLTR | 0.1 | 0.1 | 0.4 | 1.9 | 1.1 | 5.7 |
| | DetectGPT | 0.1 | 0.1 | 0.6 | 1.1 | 1.2 | 2.3 |
| Gemini Reviews | Anchor | 0.2 | 59.7 | 0.8 | **80.3** | 1.3 | **86.5** |
| | Binoculars | 0.2 | **61.5** | 0.6 | 78.0 | 1.0 | 85.5 |
| | s5 | 0.0 | 0.2 | 0.5 | 9.6 | 1.1 | 19.4 |
| | MFD | 0.1 | 0.5 | 0.4 | 8.9 | 1.1 | 18.8 |
| | FastDetectGPT | 0.1 | 1.1 | 0.5 | 5.8 | 1.1 | 10.3 |
| | GLTR | 0.2 | 0.5 | 0.8 | 5.2 | 1.8 | 12.4 |
| | DetectGPT | 0.1 | 0.4 | 0.5 | 3.5 | 1.2 | 7.0 |
| | MAGE-D | 0.1 | 0.4 | 0.6 | 3.3 | 1.3 | 7.0 |
| | Loglikelihood | 0.0 | 0.0 | 0.3 | 0.1 | 0.5 | 1.0 |
| Claude Reviews | Anchor | 0.1 | **59.6** | 0.5 | **75.8** | 1.0 | **81.8** |
| | Binoculars | 0.2 | 43.5 | 0.6 | 65.8 | 1.0 | 77.0 |
| | s5 | 0.0 | 0.1 | 0.2 | 7.6 | 0.5 | 17.5 |
| | MFD | 0.0 | 0.2 | 0.1 | 6.8 | 0.4 | 16.5 |
| | DetectGPT | 0.1 | 0.5 | 0.6 | 5.3 | 1.2 | 11.1 |
| | GLTR | 0.0 | 0.0 | 0.2 | 0.5 | 0.6 | 1.8 |

AUC for AI peer review (Figure 2) and outperforms other baseline methods overall (Table 2). The performance gap is especially notable for GPT-4o, which is generally harder to detect; for example, at a target FPR of 0.1%, the anchor embedding method achieves a TPR of 63.5% for GPT-4o reviews, compared to 17.1% for the next-best method (Binoculars), which is an absolute improvement of 46.4%. This indicates that the anchor embedding approach is particularly effective for the most challenging review samples. On Gemini reviews, where baseline methods already perform well, the anchor embedding method performs similarly to Binoculars. These results suggest that leveraging the context of the submitted manuscript through anchor embeddings offers substantial benefits for high-risk, low-FPR detection tasks, especially when reviews are written by the most advanced LLMs.

## 5 ANALYSIS

### 5.1 IS AI PEER REVIEW DETECTION ROBUST TO PROMPT VARIATIONS?

A potential limitation in constructing AI-generated datasets is the reliance on a fixed prompt, which may constrain stylistic diversity or lead to overfitting of downstream detection models. Moreover, such fixed prompting may not reflect real-world use cases, where different users are likely to employ varied and personalized prompts when generating reviews. While our primary prompt is designed to be semantically grounded—using the paper content and a human reviewer's overall score to guide generation—it is important to evaluate whether this design leads to narrow outputs or brittle detection performance. To assess the impact of prompt variation, we conducted a prompt sensitivity analysis using an alternative prompting strategy that simulates diverse reviewer archetypes (e.g., "balanced,"

"conservative," "innovative," and "nitpicky"; see Appendix E.4 for the exact prompts we used). For this analysis, we generated 1,921 GPT-4o reviews for papers corresponding to the ICLR 2021 test set.

**Cross-Prompt Detection Robustness**   We evaluated whether prompt-induced shifts in review style affect the robustness of AI text detectors. Specifically, we applied detection models calibrated on the main (i.e., score-aligned) prompt and evaluated them on reviews generated using the archetype-based prompt. Table S10 reports the TPRs at multiple target FPRs for a range of baseline detection methods. Across all settings, we observe that TPRs under cross-prompt testing remain largely consistent with those under in-prompt testing, with minimal change in detection performance. This result suggests that, despite changes in prompt framing, the core distributional properties of the AI-generated reviews are sufficiently stable for detection models to generalize. Overall, these findings indicate that the utility of our dataset is not overly sensitive to prompt selection.

**Embedding-Based Style Consistency**   We also investigated prompt-induced variation using sentence-level embedding analysis. Figure S2 visualizes a t-SNE projection of review embeddings from two groups: (i) AI-generated reviews using the primary score-aligned prompt and (ii) AI-generated reviews using the archetype prompt. No meaningful separation emerged between these two review types, suggesting they share broadly similar linguistic and semantic characteristics. This finding reinforces the robustness of our prompting strategy and implies that moderate prompt variation does not lead to drastic stylistic shifts in model outputs.

**Evaluation on Agent-Based and Structured Generation Pipelines**   To assess whether more complex prompting strategies fundamentally alter the detectability of AI-generated peer reviews, we evaluated Binoculars on reviews generated by several recent agent-based or structured LLM review systems: AI Scientist (Lu et al., 2024), AgentReview (Jin et al., 2024), and DeepReview (Zhu et al., 2025). These systems incorporate features like multi-step workflows, reviewer personas, and self-reflection. We paired these AI-generated reviews with human-written reviews from ICLR 2022 and measured detectability using AUROC (full results in Appendix D.3). Despite the increased sophistication of these generation pipelines, we found Binoculars consistently achieve high AUROC ($> 0.99$), which is ~0.03 larger than its performance on our own GPT-4o-generated baseline. This suggests that such advanced prompting or agentic strategies do not fundamentally degrade detectability, supporting the broader applicability of our benchmark.

### 5.2   CAN DETECTION MODELS DISTINGUISH HUMAN REVIEWS EDITED BY LLMs?

LLMs are widely used for writing assistance tasks such as grammar correction and fluency enhancement Laban et al. (2024); Raheja et al. (2024), especially benefiting non-native speakers. However, excessive reliance on LLMs can lead to substantial rewriting, blurring the line between human- and AI-authored reviews. To simulate this scenario, we took human-written peer reviews from the ICLR 2021 GPT-4o test subset and generated AI-edited versions at four increasing levels of editing: *Minimum*, *Moderate*, *Extensive*, and *Maximum* (see Appendix E.3 for prompts). To validate that the edits are semantically distinct, we measured cosine similarity between the original and edited reviews using a sentence embedding model. As shown in Table S14, similarity scores decrease as the level of editing increases.

We evaluate detection models on two fronts: (i) their ability to flag edited reviews as AI-generated, and (ii) their ability to rank the reviews by degree of AI involvement using Normalized Discounted Cumulative Gain (NDCG) Järvelin & Kekäläinen (2002). Table 3 compares our anchor embedding method with Binoculars, the strongest-performing baseline from previous experiments (as shown in Table 2). Both models achieve high NDCG scores (0.90 for Anchor, 0.86 for Binoculars), indicating that they can generally rank reviews according to the degree of AI editing. However, their flagging behavior differs. At the 0.1% FPR threshold, both models rarely flag minimally to extensively edited reviews as AI-generated ($\leq 2.5\%$), but diverge sharply at the Maximum level: Anchor flags 60.8% of these reviews, compared to just 9.2% for Binoculars. This trend persists at the 1% FPR threshold, with generally higher flagging rates across all levels. These results suggest that the Anchor method is more sensitive to high levels of AI involvement, whereas Binoculars is more conservative at the upper end.

Table 3: NDCG and the proportion of reviews flagged as AI-written for different levels of editing. Flagging rates are shown for two different thresholds (0.1% and 1% target FPRs), adapted from those used in Table 2.

| Method | NDCG | Threshold: 0.1%FPR) | | | | Threshold: 1%FPR) | | | |
|--------|------|---------|----------|-----------|---------|---------|----------|-----------|---------|
| | | Minimum | Moderate | Extensive | Maximum | Minimum | Moderate | Extensive | Maximum |
| Anchor | 0.90 | 0.4 | 0.7 | 1.9 | 60.8 | 2.6 | 4.9 | 9.7 | 82.3 |
| Binoculars | 0.86 | 0.6 | 1.4 | 2.5 | 9.2 | 2.7 | 5.3 | 9.4 | 26.4 |

Overall, these results suggest that high-performing detection models can distinguish varying levels of AI editing in human-written reviews. However, lightly edited texts remain difficult to detect, highlighting a challenge for future work on identifying hybrid human–AI content. Additionally, the inverse of this analysis (i.e., "human revises AI draft") could be a promising direction to investigate in future work; we expect to see an inverse relationship in this setting as our "AI edits human draft" results, where greater levels of human editing will reduce the likelihood of the review being flagged as AI-written.

## 5.3 DETECTABILITY OF MIXED-AUTHORSHIP PEER REVIEWS

In addition to investigating the detectability of human-written and LLM-edited peer reviews, we also conducted an analysis of a different mixed-authorship scenario where AI peer reviews are generated from bullet points (see Lee et al. (2024) for related approaches). Specifically, we aimed to simulate the scenario of a human drafting bullet points and using an LLM to assist with writing the full review. We did this by first asking an LLM (GPT-4.1) to condense a human-written peer review into a series of bullet points which capture the core points of the review using the following prompt: *Given a peer review for an academic research paper, condense the core points of the review into bullet points. Output only the bullet points with no additional commentary or formatting.* We then prompted GPT-4.1 to generate a review solely from the resulting bullet points (which themselves were derived from human-written reviews sampled from the ICLR 2021 test set) using the following prompt: *Given a set of bullet points describing a peer review of an academic research paper, write the full text of the peer review.*

The Anchor embedding method detection results for the 500 AI reviews generated from bullet points are provided in Table 4. Consistent with our previous analysis of mixed-authorship generation settings (AI editing of human-written peer reviews), we find that these reviews which are generated from the core points of human-written peer reviews are much more challenging to detect. This generation setting could be viewed as a legitimate use case

Table 4: Anchor detectability results for AI reviews generated from bullet points

| Target FPR | Actual FPR | Actual TPR |
|------------|------------|------------|
| 0.1% | 0.2% | 5.8% |
| 0.5% | 0.8% | 14.4% |
| 1.0% | 2.2% | 18.6% |

for LLMs in the peer review process because the core points and ideas of the review are still originating from a human, with the LLM serving as a writing assistant. Our main focus is the identification of unethical uses of LLMs in the peer review process (i.e., when it is used to fully generate the peer review); these results demonstrate how our Anchor method performs well at detecting such uses while having relatively low detection rates for cases where LLMs are primarily used for writing assistance.

## 5.4 HOW DO HUMAN-WRITTEN AND AI-GENERATED PEER REVIEWS DIFFER?

To better understand the characteristics which differentiate peer reviews written by humans and LLMs, we conducted a quantitative analysis of 32 reviews authored by humans and GPT-4o for 5 papers submitted to ICLR 2021. Specifically, we read an equal number of human and GPT-4o written reviews for each paper and noted differences in the content between them. A distinguishing characteristic of the analyzed human reviews was that they usually contained details or references to specific sections, tables, figures, or results in the paper. In contrast, peer reviews authored by GPT-4o lacked such specific details, instead focusing on higher-level comments. Another key difference identified in our qualitative analysis was the lack of any specific references to prior or related work in peer reviews generated by GPT-4o. Human-authored peer reviews often point out missing references, challenge the novelty of the paper by referencing related work, or suggest specific baselines with references that should be included in the study. In contrast, none of the analyzed GPT-4o reviews

contained such specific references to related work. Finally, we found that the vast majority of GPT-4o reviews mentioned highly similar generic criticisms which were not found in human-authored reviews for the same paper. Examples of these issues are provided in Table S12 in Appendix D.4.

Prior work has shown that peer reviews written by GPT-4 and humans have a level of semantic similarity which is comparable to that between different human-authored peer reviews, which has been used to advocate for the usefulness of feedback from GPT-4 in the paper writing process (Liang et al., 2024b). In our qualitative analysis, we found that GPT-4 does indeed generate similar higher-level comments as human reviewers, which could account for this semantic similarity. Despite being generic in nature, we would agree that such feedback could be useful to authors seeking to improve their manuscripts. Nevertheless, we believe that the lack of specificity, detail, and consideration of related work in peer reviews authored by GPT-4 demonstrates that it is not suitable for replacing human domain experts in the peer review process.

A natural concern is that the deficiencies of AI peer reviews (e.g., lack of references, generic feedback) could be artifacts of prompting, potentially affecting detectability. We tested this by generating 6,000 reviews with prompts updated to incorporate our findings above, explicitly instructing models to avoid typical AI review characteristics and to mimic more human-like reviews (Appendix E.5). Detection performance was essentially unchanged (AUROC differences $\leq 0.01$), suggesting that (1) the gap between human and AI reviews is systematic rather than prompt-dependent, and (2) our results are consistent with the prompt-robustness analysis in Section 5.1.

### 5.5 Do AI-generated reviews assign higher scores than human reviews?

In addition to qualitative differences in the content of human and AI-written reviews, we also observe a divergence in numeric scores assigned as part of the review. Figure S3 in Appendix D.6 provides histograms depicting the distribution of score differences for soundness, presentation, contribution, and confidence, which are computed by subtracting scores assigned for each category by human reviewers from those assigned by AI reviewers. AI-written peer reviews were matched with their corresponding human review (aligned by paper ID and overall recommendation) to compute the score differences. Confidence scores range from 1 to 5, while all other categories of scores range from 1 to 4. In the following discussion, We focus on reviews from NeurIPS 2022, which were produced prior to the release of ChatGPT. This provides greater confidence that the human-labeled reviews were indeed written by humans, with little to no potential AI influence.

All LLMs produce higher scores than human reviews with a high degree of statistical significance, assessed using a two-sided Wilcoxon signed-rank test (see legend for $p$-values in Figure S3). While the difference between human and AI confidence scores are relatively consistent across all three LLMs, Claude exceeds human scores by the greatest magnitude for soundness, presentation, and contribution. GPT-4o and and Gemini exceed human scores by a similar magnitude for presentation and contribution, while GPT-4o exhibits a greater divergence for soundness scores. Overall these results indicate that AI peer reviews are more favorable w.r.t. assigned scores than human-written peer reviews, which raises fairness concerns as scores are highly correlated with acceptance decisions. Our findings are consistent with prior work which has shown that papers reviewed by LLMs have a higher chance of acceptance (Drori & Te'eni, 2024; Latona et al., 2024; Ye et al., 2024).

## 6 Conclusion

In this work, we introduced a new large-scale dataset of parallel human-written and AI-generated peer reviews for identical papers submitted to leading AI research conferences. Our evaluations show that existing open-source methods for AI text detection struggle in the peer review setting, where high detection rates often come at the cost of falsely flagging human-written reviews—an outcome that must be minimized in practice. We demonstrate that leveraging manuscript context is a promising strategy for improving detection accuracy while maintaining a low false positive rate. In addition, AI-generated reviews tend to be less specific and less grounded in the manuscript than human-written ones. We also find that AI-generated reviews consistently assign higher scores, raising fairness concerns in score-driven decision-making processes. We hope our results motivate further research on responsible detection of AI-generated content in scientific review workflows, and that our dataset provides a valuable resource for advancing this goal.

## 7 Ethics Statement

Our work adheres to ethical AI principles. Peer review plays a critical role in advancing scientific discovery; however, the misuse of AI tools by reviewers to generate reviews without proper diligence can compromise the integrity of the review process. Furthermore, consistent with previous studies, we have observed that AI-generated reviews tend to be overly generic, often failing to provide actionable feedback for authors. Additionally, AI reviewers generally assign higher scores compared to human reviewers, raising concerns that AI-assisted reviews could contribute to the acceptance of work that may not meet established human evaluation standards. By developing methods to detect AI-generated reviews, our work seeks to mitigate the misuse of AI tools in peer review and promote a more rigorous and fair scientific review process.

We utilize a wide range of diverse prompts throughout our work in order to systematically evaluate how different prompting strategies may impact the detectability of AI-generated peer reviews. Nevertheless, the potential set of real-world prompts that could be used to produce AI peer reviews is vast and undefined; it is therefore impossible to construct a dataset with complete coverage of all possible prompting strategies. A user study could be valuable for further characterizing the range of prompts which could be employed in practice to produce AI-generated peer reviews. However, we note that recruiting participants for such a study presents ethical challenges because our main focus is specifically on detecting unethical applications of LLMs in the peer review process. Therefore, the target population for participants in this study would be individuals who are willing to (1) admit that they engage in unethical behavior that violates the review policies of most major conferences, and (2) are willing to share the prompts that they use to engage in such behavior with researchers who are actively working to detect it. Addressing these challenges in characterizing how humans are utilizing LLMs to write peer reviews remains a valuable direction of study for future work.

## 8 Reproducibility Statement

The full dataset is publicly released. All baselines are implemented using public codebases (IMGTB), and our Anchor method and prompting setup are described in detail in the main text and appendix.

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

# A    BASELINE METHODS

We compare our approach to 18 baseline methods from IGMBT[3] (released under MIT license) with its default setting (Spiegel & Macko, 2023), which are categorized into metric-based and pretrained model-based methods. The metric-based methods include Binoculars  Hans et al. (2024), DetectLLM-LLR  Su et al., DNAGPT  Yang et al. (2023), Entropy  Gehrmann et al. (2019), FastDetectGPT  Bao et al. (2023), GLTR  Gehrmann et al. (2019), LLMDeviation  Wu & Xiang (2023), Loglikelihood  Solaiman et al. (2019), LogRank  Mitchell et al. (2023a), MFD  Wu & Xiang (2023), Rank  Gehrmann et al. (2019), and S5  Spiegel & Macko (2023). The model-based methods include NTNU-D  Sivesind & Winje (2023), ChatGPT-D  Guo et al. (2023), OpenAI-D  Solaiman et al. (2019), OpenAI-D-lrg  Solaiman et al. (2019), RADAR-D  Solaiman et al. (2019), and MAGE-D  Li et al. (2024).

## A.1    METRIC BASED METHODS

### A.1.1    BINOCULARS

Binoculars Hans et al. (2024) analyzes text through two perspectives. First, it calculates the log perplexity of the text using an observer LLM. Then, a performer LLM generates next-token predictions, whose perplexity is evaluated by the observer—this metric is termed cross-perplexity. The ratio of perplexity to cross-perplexity serves as an indicator for detecting LLM-generated text.

### A.1.2    DNAGPT

DNAGPT Yang et al. (2023) is a training-free detection method designed to identify machine-generated text. Unlike conventional approaches that rely on training models, DNAGPT uses Divergent N-Gram Analysis (DNA) to detect discrepancies in text origin. The method works by truncating a given text at the midpoint and using the preceding portion as input to an LLM to regenerate the missing section. By comparing the regenerated text with the original through N-gram analysis (black-box) or probability divergence (white-box), DNAGPT reveals distributional differences between human and machine-written text, offering a flexible and explainable detection strategy.

### A.1.3    ENTROPY

Similar to the Rank score, the Entropy score for a text is determined by averaging the entropy values of each word, conditioned on its preceding context  Gehrmann et al. (2019).

### A.1.4    GLTR

The Entropy score, like the Rank score, is computed by averaging the entropy values of each word within a text, considering the preceding context  Gehrmann et al. (2019).

### A.1.5    MFD

The Multi-Feature Detection (MFD) method Wu & Xiang (2023) detects AI-generated text using four features: log-likelihood, log-rank, entropy, and LLM deviation.

### A.1.6    LOGLIKELIHOOD

This method utilizes a language model to compute the token-wise log probability. Specifically, given a text, the log probability of each token is averaged to produce a final score. A higher score indicates a greater likelihood that the text is machine-generated  Solaiman et al. (2019).

### A.1.7    LOGRANK

Unlike the Rank metric, which relies on absolute rank values, the Log-Rank score is derived by applying a logarithmic function to the rank value of each word  Mitchell et al. (2023a).

---

[3]`https://github.com/kinit-sk/IMGTB`

### A.1.8 RANK

The Rank score is calculated by determining the absolute rank of each word in a text based on its preceding context. The final score is obtained by averaging the rank values across the text. A lower score suggests a higher probability that the text was machine-generated Gehrmann et al. (2019).

### A.1.9 DETECTLLM-LLR

This approach integrates Log-Likelihood and Log-Rank scores, leveraging their complementary properties to analyze a given text Su et al..

### A.1.10 FASTDETECTGPT

This method assesses changes in a model's log probability function when small perturbations are introduced to a text. The underlying idea is that LLM-generated text often resides in a local optimum of the model's probability function. Consequently, minor perturbations to machine-generated text typically result in lower log probabilities, whereas perturbations to human-written text may lead to either an increase or decrease in log probability Mitchell et al. (2023a).

### A.2 MODEL-BASED METHODS

### A.2.1 NTNU-D

It is a fine-tuned classification model based on the RoBERTa-base model, and three sizes of the bloomz-models Sivesind & Winje (2023)

### A.2.2 CHATGPT-D

The ChatGPT Detector Guo et al. (2023) is designed to differentiate between human-written text and content generated by ChatGPT. It is based on a RoBERTa model that has been fine-tuned for this specific task. The authors propose two training approaches: one that trains the model solely on generated responses and another that incorporates both question-answer pairs for joint training. In our evaluation, we adopt the first approach to maintain consistency with other detection methods.

### A.2.3 OPENAI-D AND RADAR-D

OpenAI Detector Solaiman et al. (2019) is a fine-tuned RoBERTa model designed to identify outputs generated by GPT-2. Specifically, it was trained using text generated by the largest GPT-2 model (1.5B parameters) and is capable of determining whether a given text is machine-generated.

### A.2.4 MAGE-D

MAGE (MAchine-GEnerated text detection) Li et al. (2024) is a large-scale benchmark designed for detecting AI-generated text. It compiles human-written content from seven diverse writing tasks, including story generation, news writing, and scientific writing. Corresponding machine-generated texts are produced using 27 different LLMs, such as ChatGPT, LLaMA, and Bloom, across three representative prompt types.

## B DATASET DETAILS

### B.1 DATASET

A subset of our dataset is included as part of the supplementary materials in a zipped file. The full dataset will be released publicly upon publication under a permissive research-use license.

### B.2 DATASET FILE STRUCTURE

The calibration, test, and extended sets are in separate directories. Each directory contains subdirectories for different models that were used to generate AI peer review samples. In each model's subdi-

rectory, you will find multiple CSV files, with each file representing peer review samples of a specific conference. Each file follows the naming convention: "`<conference>.<subset>.<LLM>.csv`". The directory and file structure are outlined below.

```
|-- calibration
    |-- gpt4o
        |-- ICLR2017.calibration.gpt-4o.csv
        |-- ...
        |-- ICLR2024.calibration.gpt-4o.csv
        |-- NeurIPS2016.calibration.gpt-4o.csv
        |-- ...
        |-- NeurIPS2024.calibration.gpt-4o.csv
    |-- claude
        |-- ...
    |-- gemini
        |-- ...
    |-- llama
        |-- ...
    |-- qwen
        |-- ...
|-- extended
    |-- gpt4o
        |-- ICLR2018.extended.gpt-4o.csv
        |-- ...
        |-- ICLR2024.extended.gpt-4o.csv
        |-- NeurIPS2016.extended.gpt-4o.csv
        |-- ...
        |-- NeurIPS2024.extended.gpt-4o.csv
    |-- llama
        |-- ...
|-- test
    |-- gpt4o
        |-- ICLR2017.test.gpt-4o.csv
        |-- ...
        |-- ICLR2024.test.gpt-4o.csv
        |-- NeurIPS2016.test.gpt-4o.csv
        |-- ...
        |-- NeurIPS2024.test.gpt-4o.csv
    |-- claude
        |-- ...
    |-- gemini
        |-- ...
    |-- llama
        |-- ...
    |-- qwen
        |-- ...
```

### B.3  CSV File Content

CSV files may differ in their column structures across conferences and years. These differences are due to updates in the required review fields over time as well as variations between conferences. See Table S1 for review fields of individual conferences.

Table S1: Required fields in the review templates for each conference.

| Conference | Required Fields |
|---|---|
| ICLR2017 | review, rating, confidence |
| ICLR2018 | review, rating, confidence |
| ICLR2019 | review, rating, confidence |
| ICLR2020 | review, rating, confidence, experience assessment, checking correctness of derivations and theory, checking correctness of experiments, thoroughness in paper reading |
| ICLR2021 | review, rating, confidence |
| ICLR2022 | summary of the paper, main review, summary of the review, correctness, technical novelty and significance, empirical novelty and significance, flag for ethics review, recommendation, confidence |
| ICLR2023 | summary of the paper, strength and weaknesses, clarity quality novelty and reproducibility, summary of the review, rating, confidence |
| ICLR2024 | summary, strengths, weaknesses, questions, soundness, presentation, contribution, flag for ethics review, rating, confidence |
| NeurIPS2016 | review, rating, confidence |
| NeurIPS2017 | review, rating, confidence |
| NeurIPS2018 | review, overall score, confidence score |
| NeurIPS2019 | review, overall score, confidence score, contribution |
| NeurIPS2021 | summary, main review, limitations and societal impact, rating, confidence, needs ethics review, ethics review area |
| NeurIPS2022 | summary, strengths and weaknesses, questions, limitations, ethics flag, ethics review area, rating, confidence, soundness, presentation, contribution |
| NeurIPS2023 | summary, strengths, weaknesses, questions, limitations, ethics flag, ethics review area, rating, confidence, soundness, presentation, contribution |
| NeurIPS2024 | summary, strengths, weaknesses, questions, limitations, ethics flag, ethics review area, rating, confidence, soundness, presentation, contribution |

## B.4 Dataset Sample Numbers per Conference Year

In this section, we present further breakdowns of sample numbers by conference, year, and LLM, as shown in Table 1.

Table S2: Entire set sample size, including both human and AI reviews. They are exactly balanced.

| Conference | gpt4o | llama |
|---|---|---|
| ICLR2017 | 2926 | 2918 |
| ICLR2018 | 5460 | 5434 |
| ICLR2019 | 9414 | 9378 |
| ICLR2020 | 15426 | 15366 |
| ICLR2021 | 18786 | 18768 |
| ICLR2022 | 20042 | 20026 |
| ICLR2023 | 28562 | 28560 |
| ICLR2024 | 55714 | 55672 |
| NeurIPS2016 | 6296 | 6284 |
| NeurIPS2017 | 3848 | 3774 |
| NeurIPS2018 | 5990 | 5938 |
| NeurIPS2019 | 8444 | 8398 |
| NeurIPS2021 | 21170 | 21164 |
| NeurIPS2022 | 20472 | 20408 |
| NeurIPS2023 | 30264 | 30194 |
| NeurIPS2024 | 33206 | 33104 |

Table S3: Test set sample size, including both human and AI reviews. They are exactly balanced.

| Conference | gemini | claude | qwen | gpt4o | llama |
|---|---|---|---|---|---|
| ICLR2017 | 2924 | 2926 | 2918 | 2926 | 2918 |
| ICLR2018 | 3000 | 3004 | 2988 | 3004 | 2992 |
| ICLR2019 | 3002 | 3010 | 3000 | 3010 | 2998 |
| ICLR2020 | 3016 | 3022 | 3000 | 3022 | 3010 |
| ICLR2021 | 3840 | 3842 | 3830 | 3842 | 3838 |
| ICLR2022 | 3896 | 3900 | 3838 | 3900 | 3898 |
| ICLR2023 | 3816 | 3816 | 3816 | 3816 | 3814 |
| ICLR2024 | 3784 | 3820 | 3800 | 3822 | 3816 |
| NeurIPS2016 | 5522 | 5534 | 5534 | 5536 | 5526 |
| NeurIPS2017 | 2854 | 2858 | 2850 | 2858 | 2812 |
| NeurIPS2018 | 3000 | 3006 | 2916 | 3006 | 2982 |
| NeurIPS2019 | 2930 | 2938 | 2922 | 2940 | 2928 |
| NeurIPS2021 | 3884 | 3884 | 3884 | 3884 | 3884 |
| NeurIPS2022 | 3606 | 3622 | 3598 | 3622 | 3610 |
| NeurIPS2023 | 4436 | 4440 | 4382 | 4440 | 4432 |
| NeurIPS2024 | 3914 | 3926 | 3880 | 3926 | 3912 |

Table S4: Calibration set sample size, including both human and AI reviews. They are exactly balanced.

| Conference | gemini | claude | qwen | gpt4o | llama |
|---|---|---|---|---|---|
| ICLR2021 | 3826 | 3828 | 3802 | 3828 | 3828 |
| ICLR2022 | 3878 | 3844 | 3860 | 3882 | 3880 |
| NeurIPS2021 | 3828 | 3830 | 3818 | 3830 | 3828 |
| NeurIPS2022 | 3648 | 3654 | 3634 | 3654 | 3644 |

## C  ADDITIONAL RESULTS

### C.1  UNCERTAINTY ESTIMATES VIA BOOTSTRAP RESAMPLING

To estimate the uncertainty of our main experimental results (TPR and FPR reported in Table 2), we perform bootstrap resampling with replacement using $N = 100$ resamples. For each method and evaluation setting, we compute the standard deviation (SD) of the TPR and FPR across the 100 bootstrap replicates. While 100 resamples provide only a coarse estimate of variability, this level of resampling was chosen to balance computational cost with the need to quantify uncertainty. As shown in Table S5, the standard deviations are small relative to the large performance gaps between methods, indicating the robustness of our main findings.

### C.2  CALIBRATION USING ICLR + NEURIPS REVIEWS

Our main results in Table 2 of Section 4.2 utilized ICLR review from our calibration set to calibrate each detection method. This simulates the scenario in which some of the reviews in the test set are "out-of-domain" in the sense that they belong to a different conference than the reviews used for calibration. In Table S6, we provide additional results for the same evaluation setting as before, but using both ICLR and NeurIPS reviews for calibration (i.e., fully "in domain"). We generally see similar trends regarding relative performance between methods as before, with the exception that the Binoculars method achieves slightly higher detection rates than our Anchor method for Gemini reviews. This suggests that existing methods such as Binoculars may be more sensitive to the use of in-domain data during calibration.

Table S5: Actual true positive rate (TPR) and false positive rate (FPR), ± their standard deviation, computed over 100 bootstrap resamples corresponding to the main results in Table 2.

| | Target FPR: | 0.1% | | 0.5% | | 1% | |
|---|---|---|---|---|---|---|---|
| | | FPR | TPR | FPR | TPR | FPR | TPR |
| **GPT-4o Reviews** | Anchor | 0.1 ± 0.02 | **63.5** ± 0.34 | 0.5 ± 0.03 | **83.7** ± 0.33 | 1.0 ± 0.04 | **88.8** ± 0.31 |
| | Binoculars | 0.2 ± 0.03 | 17.1 ± 0.25 | 0.6 ± 0.05 | 33.6 ± 0.31 | 1.0 ± 0.06 | 45.2 ± 0.34 |
| | MAGE-D | 0.1 ± 0.03 | 2.3 ± 0.11 | 0.6 ± 0.05 | 8.8 ± 0.21 | 1.3 ± 0.07 | 14.7 ± 0.24 |
| | s5 | 0.1 ± 0.03 | 0.1 ± 0.02 | 0.9 ± 0.07 | 7.2 ± 0.20 | 1.7 ± 0.09 | 17.5 ± 0.29 |
| | MFD | 0.2 ± 0.03 | 0.1 ± 0.02 | 0.8 ± 0.07 | 6.0 ± 0.15 | 1.6 ± 0.09 | 15.6 ± 0.24 |
| | GLTR | 0.1 ± 0.02 | 0.1 ± 0.02 | 0.4 ± 0.05 | 1.9 ± 0.08 | 1.1 ± 0.08 | 5.7 ± 0.14 |
| | DetectGPT | 0.1 ± 0.02 | 0.1 ± 0.02 | 0.6 ± 0.05 | 1.1 ± 0.07 | 1.2 ± 0.08 | 2.3 ± 0.10 |
| **Gemini Reviews** | Anchor | 0.2 ± 0.01 | 59.7 ± 0.32 | 0.8 ± 0.03 | **80.3** ± 0.33 | 1.3 ± 0.04 | **86.5** ± 0.31 |
| | Binoculars | 0.2 ± 0.03 | **61.5** ± 0.32 | 0.6 ± 0.05 | 78.0 ± 0.28 | 1.0 ± 0.07 | 85.5 ± 0.23 |
| | s5 | 0.0 ± 0.01 | 0.2 ± 0.03 | 0.5 ± 0.04 | 9.6 ± 0.21 | 1.1 ± 0.08 | 19.4 ± 0.27 |
| | MFD | 0.1 ± 0.02 | 0.5 ± 0.04 | 0.4 ± 0.05 | 8.9 ± 0.17 | 1.1 ± 0.07 | 18.8 ± 0.25 |
| | FastDetectGPT | 0.1 ± 0.02 | 1.1 ± 0.08 | 0.5 ± 0.04 | 5.8 ± 0.18 | 1.1 ± 0.06 | 10.3 ± 0.21 |
| | GLTR | 0.2 ± 0.03 | 0.5 ± 0.04 | 0.8 ± 0.06 | 5.2 ± 0.13 | 1.8 ± 0.09 | 12.4 ± 0.23 |
| | DetectGPT | 0.1 ± 0.02 | 0.4 ± 0.05 | 0.5 ± 0.05 | 3.5 ± 0.12 | 1.2 ± 0.08 | 7.0 ± 0.18 |
| | MAGE-D | 0.1 ± 0.03 | 0.4 ± 0.04 | 0.6 ± 0.06 | 3.3 ± 0.12 | 1.3 ± 0.08 | 7.0 ± 0.17 |
| | Loglikelihood | 0.0 ± 0.01 | 0.0 ± 0.00 | 0.3 ± 0.04 | 0.1 ± 0.02 | 0.5 ± 0.05 | 1.0 ± 0.07 |
| **Claude Reviews** | Anchor | 0.1 ± 0.02 | **59.6** ± 0.34 | 0.5 ± 0.04 | **75.8** ± 0.34 | 1.0 ± 0.05 | **81.8** ± 0.32 |
| | Binoculars | 0.2 ± 0.04 | 43.5 ± 0.33 | 0.6 ± 0.05 | 65.8 ± 0.33 | 1.0 ± 0.07 | 77.0 ± 0.29 |
| | s5 | 0.0 ± 0.01 | 0.1 ± 0.02 | 0.2 ± 0.03 | 7.6 ± 0.18 | 0.5 ± 0.05 | 17.5 ± 0.23 |
| | MFD | 0.0 ± 0.01 | 0.2 ± 0.03 | 0.1 ± 0.03 | 6.8 ± 0.18 | 0.4 ± 0.05 | 16.5 ± 0.24 |
| | DetectGPT | 0.1 ± 0.02 | 0.5 ± 0.05 | 0.6 ± 0.05 | 5.3 ± 0.16 | 1.2 ± 0.06 | 11.1 ± 0.22 |
| | GLTR | 0.0 ± 0.01 | 0.0 ± 0.01 | 0.2 ± 0.03 | 0.5 ± 0.05 | 0.6 ± 0.05 | 1.8 ± 0.10 |

Table S6: Actual FPR and TPR calculated from the withheld test dataset at varying detection thresholds, which are calibrated using ICLR and NeurIPS reviews from our calibration set at different target FPRs. Best TPRs are in **bold**.

| | Target FPR: | 0.1% | | 0.5% | | 1% | |
|---|---|---|---|---|---|---|---|
| | | FPR | TPR | FPR | TPR | FPR | TPR |
| GPT-4o Reviews | Anchor | 0.1 | **61.4** | 0.3 | **80.1** | 0.8 | **87.4** |
| | Binoculars | 0.3 | 18.8 | 0.7 | 37.5 | 1.2 | 49.3 |
| | MAGE-D | 0.1 | 2.3 | 0.7 | 9.6 | 1.3 | 14.5 |
| | s5 | 0.3 | 0.7 | 1.0 | 8.0 | 1.6 | 16.3 |
| | MFD | 0.3 | 0.9 | 0.9 | 7.8 | 1.6 | 14.9 |
| | GLTR | 0.1 | 0.1 | 0.5 | 2.4 | 1.2 | 5.9 |
| | DetectGPT | 0.1 | 0.2 | 0.6 | 1.1 | 1.0 | 2.1 |
| | Loglikelihood | 0.1 | 0.0 | 0.3 | 0.2 | 0.6 | 1.0 |
| Gemini Reviews | Binoculars | 0.3 | **63.8** | 0.7 | **80.9** | 1.1 | **87.6** |
| | Anchor | 0.2 | 57.2 | 0.5 | 75.5 | 1.1 | 84.2 |
| | MFD | 0.1 | 1.9 | 0.6 | 11.0 | 1.0 | 18.1 |
| | s5 | 0.1 | 1.4 | 0.5 | 10.5 | 1.0 | 18.3 |
| | GLTR | 0.2 | 0.6 | 1.0 | 6.3 | 1.8 | 12.9 |
| | FastDetectGPT | 0.1 | 1.1 | 0.4 | 4.9 | 0.9 | 8.9 |
| | MAGE-D | 0.1 | 0.4 | 0.7 | 3.8 | 1.3 | 6.9 |
| | DetectGPT | 0.1 | 0.5 | 0.5 | 3.2 | 1.1 | 6.3 |
| | Loglikelihood | 0.1 | 0.0 | 0.3 | 0.2 | 0.6 | 1.6 |
| | NTNU-D | 11.5 | 0.0 | 21.8 | 0.0 | 26.3 | 0.1 |
| Claude Reviews | Anchor | 0.1 | **53.8** | 0.3 | **72.6** | 0.8 | **80.0** |
| | Binoculars | 0.3 | 46.4 | 0.7 | 70.2 | 1.1 | **80.0** |
| | MFD | 0.0 | 1.0 | 0.2 | 8.7 | 0.4 | 15.9 |
| | s5 | 0.0 | 0.7 | 0.2 | 8.4 | 0.4 | 16.4 |
| | DetectGPT | 0.1 | 0.6 | 0.5 | 4.9 | 1.0 | 10.1 |
| | GLTR | 0.0 | 0.0 | 0.3 | 0.7 | 0.6 | 1.9 |

## C.3 Additional Llama and Qwen Detection Results

Table S7 is organized similarly to Table 2, but presents results for Llama and Qwen reviews. Both tables use the same set of thresholds for each method, which were calibrated using ICLR reviews generated by GPT-4o, Gemini, and Claude along with their matching human-written reviews.

Reviews generated by open-source LLMs (Table S7) show somewhat different trends compared to reviews generated by commercial LLMs (Table 2). For both Llama and Qwen reviews, the Binoculars method achieves near-perfect detection performance. While the Anchor method ranks second for Qwen reviews, it is not as performant on Llama reviews. This is surprising, as Llama reviews appear easier to detect. For instance, at a target FPR of 0.1%, more than 6.5 times as many methods achieve a TPR above 10% compared to GPT-4o, Gemini, and Claude reviews, and about twice as many compared to Qwen reviews. One possibility is that the semantic similarity between the three anchor reviews generated by higher-quality LLMs (GPT-4o, Gemini, and Claude) and Llama reviews is low enough to overlap with that between the anchor reviews and human reviews, potentially blurring the decision boundary. Although the underlying causes of these differences warrant further investigation, these findings are less central to our study since most LLM users are likely to rely on commercial models due to their ease of use and superior capabilities.

Table S7: Actual FPR and TPR calculated from the withheld test dataset at varying detection thresholds, which are calibrated using ICLR reviews from our calibration set at different target FPRs. Best TPRs are in **bold**. Detection methods are ordered by their TPR at a target FPR of 0.1%, and those that failed to achieve 10% TPR at a target FPR of 1% are omitted.

| | Target FPR: | 0.1% | | 0.5% | | 1% | |
|---|---|---|---|---|---|---|---|
| | | FPR | TPR | FPR | TPR | FPR | TPR |
| Llama Reviews | Binoculars | 0.2 | **98.4** | 0.6 | **99.0** | 1.0 | **99.2** |
| | MAGE-D | 0.1 | 63.8 | 0.7 | 91.8 | 1.3 | 95.7 |
| | MFD | 0.0 | 57.6 | 0.1 | 81.9 | 0.1 | 87.4 |
| | GLTR | 0.1 | 55.4 | 0.2 | 75.2 | 0.3 | 81.7 |
| | FastDetectGPT | 0.1 | 54.7 | 0.5 | 73.8 | 1.2 | 80.6 |
| | s5 | 0.0 | 53.5 | 0.1 | 83.2 | 0.1 | 88.0 |
| | OpenAI-D | 0.2 | 38.8 | 0.6 | 48.3 | 1.6 | 57.1 |
| | LLMDeviation | 0.0 | 37.0 | 0.0 | 37.0 | 0.0 | 37.0 |
| | ChatGPT-D | 0.0 | 26.4 | 0.0 | 26.4 | 0.0 | 26.4 |
| | DetectLLM-LLR | 0.0 | 19.3 | 0.0 | 19.3 | 0.0 | 19.3 |
| | LogRank | 0.0 | 18.2 | 0.0 | 18.2 | 0.0 | 18.2 |
| | Loglikelihood | 0.0 | 13.4 | 0.3 | 76.0 | 0.5 | 85.5 |
| | Anchor | 0.1 | 13.6 | 0.5 | 28.5 | 1.0 | 36.8 |
| | DetectGPT | 0.1 | 1.7 | 0.7 | 10.8 | 1.3 | 19.7 |
| | RADAR-D | 0.9 | 0.0 | 2.6 | 1.6 | 4.2 | 12.5 |
| Qwen Reviews | Binoculars | 0.2 | **99.4** | 0.6 | **99.8** | 1.0 | **99.9** |
| | Anchor | 0.2 | 72.5 | 0.8 | 88.4 | 1.3 | 92.1 |
| | FastDetectGPT | 0.1 | 54.3 | 0.5 | 77.6 | 1.1 | 85.4 |
| | MFD | 0.1 | 37.6 | 0.4 | 73.0 | 0.6 | 82.3 |
| | s5 | 0.1 | 34.7 | 0.4 | 76.0 | 0.6 | 83.9 |
| | MAGE-D | 0.1 | 33.2 | 0.7 | 73.6 | 1.3 | 86.2 |
| | GLTR | 0.1 | 31.1 | 0.3 | 64.7 | 0.6 | 77.4 |
| | Loglikelihood | 0.0 | 0.5 | 0.3 | 30.8 | 0.5 | 56.6 |

## C.4 Experimental Results for Full Dataset

We test existing AI text generation text detection models on our entire dataset (i.e., the test set + the extended set). Results are provided in Tables S8 and S9 for ICLR and NeurIPS reviews (rspeectively).

Table S8: Actual FPR and TPR calculated from the ICLR reviews at varying detection thresholds, which are calibrated using the ICLR calibration dataset at different target FPRs.

| | Target FPR: | 0.1% | | 0.5% | | 1% | |
|---|---|---|---|---|---|---|---|
| | | FPR | TPR | FPR | TPR | FPR | TPR |
| GPT | Binoculars | 0.8% | 23.4% | 1.5% | 41.7% | 2.2% | 54.2% |
| | GLTR | 0.7% | 3.3% | 2.1% | 12.0% | 3.8% | 21.0% |
| Llama | Binoculars | 0.8% | 98.9% | 1.5% | 99.4% | 2.2% | 99.6% |
| | GLTR | 0.4% | 80.6% | 1.0% | 92.5% | 1.6% | 95.3% |

Table S9: Actual FPR and TPR calculated from the NeurIPS reviews at varying detection thresholds, which are calibrated using the ICLR calibration dataset at different target FPRs.

| | Target FPR: | 0.1% | | 0.5% | | 1% | |
|---|---|---|---|---|---|---|---|
| | | FPR | TPR | FPR | TPR | FPR | TPR |
| GPT | Binoculars | 0.3% | 26.1% | 0.6% | 45.3% | 0.9% | 54.1% |
| | GLTR | 0.3% | 1.3% | 5.1% | 31.9% | 10.6% | 52.6% |
| Llama | Binoculars | 0.3% | 98.9% | 0.6% | 99.3% | 0.9% | 99.4% |
| | GLTR | 0.3% | 76.3% | 0.9% | 91.0% | 1.4% | 93.9% |

## C.5 COMPUTATION TIME

We report the computation time of each detection method. Each method was evaluated on 100 samples using a single NVIDIA RTX A6000 GPU, repeated 20 times to compute the mean and standard deviation. The Anchor method is an exception: it does not use a GPU and relies on sequential API calls to the OpenAI service, which could potentially be optimized for faster execution (e.g., parallelizing API calls). Figure S1 summarizes the results.

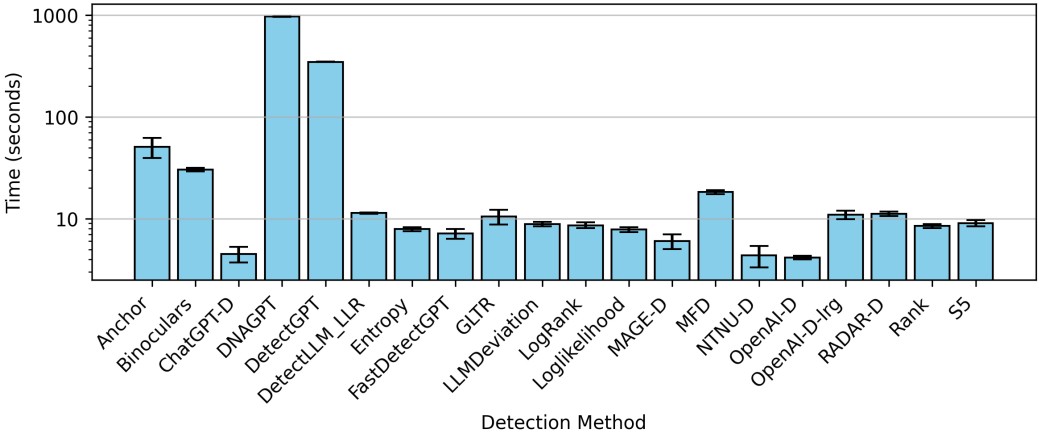

Figure S1: Computation time (in seconds) for processing 100 samples. Each method was repeated 20 times to compute the mean and standard deviation. All methods were run on a single NVIDIA RTX A6000 GPU, except for Anchor, which used sequential API calls without GPU acceleration.

# D  ADDITIONAL ANALYSES

## D.1  VOTING MECHANISM FOR THE ANCHOR EMBEDDING METHOD

Intuitively, the anchor approach performs best when the anchor embeddings are generated using the same model that produced the test review (source LLM). However, in real-world scenarios, the source LLM is typically unknown (a situation commonly referred to as "black-box" detection scenario). To address this challenge, we propose a voting-based technique. Specifically, we generate multiple anchor embeddings using different types of LLMs (anchor LLMs). For each anchor embedding, we compute the Score (Eq.4) and derive the corresponding label assignment (Eq.5). If at least one anchor embedding assigns a positive label, the final label is positive. Otherwise, the final label is negative. In our experiments, we used three anchor reviews for voting—each generated by GPT-4o, Gemini, and Claude, respectively.

## D.2  ASSESSING DATASET ROBUSTNESS TO PROMPT VARIATION

This appendix provides supporting materials for the prompt sensitivity analysis described in Section 5.1 of the main text. The results presented here include quantitative evaluations of detection performance under prompt variation (Table S10) and a visualization of review embeddings to assess stylistic consistency across prompt types (Figure S2). Together, these results support the conclusion that our prompting strategy yields stable and generalizable outputs across different prompt formulations.

Table S10: Actual FPR and TPR on the withheld ICLR2021 test set. The upper section shows results for reviews generated using the score-aligned prompt (used in our main dataset), and the lower section shows results for reviews generated using the alternative, archetype-based prompt. In both cases, the same set of thresholds, calibrated on the calibration set, is applied. Detection performance remains largely consistent across prompt styles, indicating that prompt variation does not substantially degrade model effectiveness.

| | Target FPR: | 0.1% | | 0.5% | | 1% | |
|---|---|---|---|---|---|---|---|
| | | FPR | TPR | FPR | TPR | FPR | TPR |
| Score-aligned | Anchor | 0.2 | 86.9 | 0.8 | 96.4 | 1.6 | 98.1 |
| | Binoculars | 0.1 | 7.6 | 0.3 | 21.9 | 0.6 | 32.6 |
| | s5 | 0.1 | 0.0 | 0.7 | 3.8 | 1.6 | 12.5 |
| | MFD | 0.1 | 0.0 | 0.8 | 2.8 | 1.5 | 10.8 |
| | MAGE-D | 0.0 | 0.0 | 0.1 | 1.1 | 0.3 | 2.4 |
| | GLTR | 0.1 | 0.0 | 0.4 | 0.3 | 1.1 | 1.8 |
| Archetype | Anchor | 0.2 | 83.4 | 0.8 | 94.8 | 1.6 | 96.8 |
| | Binoculars | 0.1 | 7.8 | 0.3 | 18.1 | 0.6 | 27.5 |
| | s5 | 0.1 | 0.0 | 0.8 | 6.2 | 1.6 | 14.9 |
| | MFD | 0.1 | 0.0 | 0.8 | 4.5 | 1.7 | 12.8 |
| | MAGE-D | 0.0 | 0.1 | 0.1 | 0.6 | 0.3 | 1.7 |
| | GLTR | 0.1 | 0.0 | 0.3 | 0.0 | 0.8 | 0.8 |

## D.3  ASSESSING DATASET ROBUSTNESS TO AGENT-BASED AND STRUCTURED GENERATION PIPELINES

To assess whether AI-generated reviews produced via more complex prompting pipelines differ in detectability, we evaluated Binoculars on reviews generated by recent agent-based and structured LLM systems. These include:

- **AI Scientist** (Lu et al., 2024): Combines self-reflection, ensembling, and few-shot prompting to generate reviews with both simple and complex reasoning flows.

- **AgentReview** (Jin et al., 2024): Uses GPT-4 agents simulating different reviewer archetypes (e.g., knowledgeable, irresponsible) in multi-turn interactions.

- **DeepReview** (Zhu et al., 2025): Uses a structured, multi-stage pipeline with separate modules for scoring, justification, and recommendations.

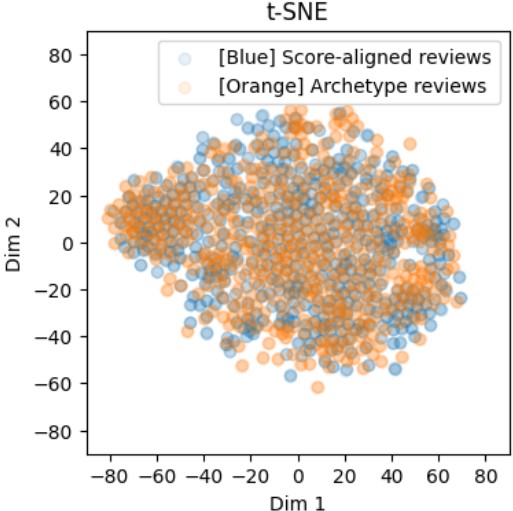

Figure S2: t-SNE visualization of sentence embeddings from AI-generated reviews in the ICLR2021 test set. Blue points represent reviews generated using the main score-aligned prompt, and orange points represent those from the alternative archetype-based prompt. The substantial overlap between the two distributions suggests that prompt variation does not cause major shifts in model outputs. Embeddings were computed using OpenAI's text-embedding-3-small model; t-SNE was performed with 2 output dimensions and a perplexity of 30.

These datasets only contain LLM-generated reviews. To compare them against human-written reviews, we paired them with our ICLR 2022 human review samples. Due to resource constraints and to minimize experimental overhead, we focus this test on Binoculars, the top-performing baseline amongst the existing AI text detection methods. As a control, we included samples from our own GPT-4o-generated reviews (ICLR 2022).

Table S11: AUROC of Binoculars on reviews from agent-based and structured LLM pipelines, paired with ICLR 2022 human-written reviews. The control row includes GPT-4o-generated reviews from our dataset for the same year. Variants of each review generation pipeline are indicated in parentheses. Mean, minimum, and maximum values are computed via bootstrapping (n=1,000).

| Source | Mean AUROC | Min | Max |
|---|---|---|---|
| AI Scientist (simple) | 0.9955 | 0.9936 | 0.9975 |
| AI Scientist (complex) | 0.9973 | 0.9933 | 0.9992 |
| AgentReview (baseline) | 0.9953 | 0.9935 | 0.9964 |
| AgentReview (benign) | 0.9984 | 0.9974 | 0.9990 |
| AgentReview (responsible) | 0.9983 | 0.9976 | 0.9991 |
| AgentReview (irresponsible) | 0.9984 | 0.9975 | 0.9991 |
| AgentReview (knowledgeable) | 0.9982 | 0.9974 | 0.9990 |
| AgentReview (unknowledgeable) | 0.9985 | 0.9977 | 0.9991 |
| DeepReview (fast) | 0.9937 | 0.9918 | 0.9954 |
| DeepReview (best) | 0.9929 | 0.9914 | 0.9950 |
| Control (GPT-4o) | 0.9647 | 0.9604 | 0.9686 |

Across all agentic and structured generation pipelines tested, Binoculars consistently achieved high AUROC — often outperforming its results on our own GPT-4o samples (Table S11). This suggests that the detectability of AI-generated peer reviews is not fundamentally diminished by the use of agent-based workflows, CoT prompting, or structured review pipelines. While we limit this evaluation to one method due to computational constraints, the results strengthen the conclusion that our benchmark remains robust and relevant, even as more complex AI reviewing systems emerge.

D.4    EXAMPLES FROM HUMAN ANALYSIS OF DIFFERENCES BETWEEN HUMAN AND AI-WRITTEN PEER REVIEWS

Table S12 provides examples of the issues identified in our qualitative analysis of human and AI-written peer reviews. In general, we observe that GPT-4o reviews lack references to specific details in the paper, lack references to specific prior work, and contain overly generic criticisms. See Section 5.4 for additional discussion.

Table S12: Examples of differences identified in human analysis of human and AI-written peer reviews

| Category | Human review example | GPT-4o review examples |
|---|---|---|
| References to specific details in the paper | "Table 2 confirms that MDR outperforms Graph Rec Retriever (Asai et al.). This result shows the feasibility of a more accurate multi-hop QA model without external knowledge such as Wikipedia hyperlinks." | "The paper extensively evaluates on multiple datasets and situates the contributions clearly within existing literature, substantiating claims with thorough quantitative analysis." |
| Specific references to prior work | "My only serious concern is the degree of novelty with respect to (Yuan et al., 2020), which was published at ECCV 2020. The main difference seems to be that in the proposed method the graph is dynamic (i.e., it depends on the input sentences), instead in (Yuan et al., 2018) the graph is learned but fixed for all the input samples." | "The novelty of the TDM is not strong enough relative to prior work." |
| Generic criticisms | N/A | "Lack of clarity" (without pointing to specific statements in the paper which need clarification); "lack of discussion of limitations or computational considerations"; "need more discussion of hyperparameter sensitivity"; "need comparisons to more datasets" (without suggesting any in particular); "technical language used in the paper may be difficult to follow for unfamiliar readers" |

D.5    PROMPT ENRICHMENT EXPERIMENT

In Section 5.4, we identified several characteristic deficiencies of AI peer reviews compared to human-written ones, such as a lack of references to specific details in the paper, limited engagement with related work, and reliance on generic or boilerplate phrasing. One natural question is whether whether these deficiencies could be artifacts of the original prompting strategy, and whether updating the prompts to explicitly address them might close the gap with human-authored reviews. To test this, we added an extra prompt to the original one that incorporated our findings from Section 5.4, instructing models to avoid these AI-like characteristics and to produce reviews that more closely resemble human ones. The following is the added prompt:

```
Do the following:
  - Reference specific elements from the paper (e.g., particular sections, tables,
  figures, equations, or experimental results).
  - Engage with related work: mention missing citations, suggest relevant prior
  studies, and evaluate the paper's novelty in the context of existing literature.
  - Provide paper-specific, varied feedback rather than relying on template-like
  language.
  - Balance high-level assessments with detailed, grounded observations.
```

```
Do NOT do the following:
  - Do not give only vague or generic comments without specific evidence from the
  paper.
  - Do not recycle identical phrasing or boilerplate critiques across reviews.
  - Do not omit discussion of related work or novelty considerations.
  - Do not provide feedback that could apply equally to almost any paper; make each
  review unique and tailored.
```

Using this revised prompt, we generated 1,000 reviews for papers in our calibration set across four conference-year pairs (ICLR and NeurIPS 2021–2022) using the same three commercial LLMs from our study (GPT-4o, Gemini, Claude). With matching human reviews, this produced a total of 6,000 reviews. We then compared the detection performance of the top two performing methods (Anchor and Binoculars) between the original and enriched prompt settings.

Table S13: Detection performance (AUROC) under original and enriched prompting strategies.

| Detector | Original Prompt | Enriched Prompt |
|---|---|---|
| Anchor | 0.995 | 0.997 |
| Binoculars | 0.982 | 0.977 |

As shown in Table S13, detection performance remained virtually unchanged (AUROC differences within 0.01), indicating that (1) the distinction between human- and AI-written reviews reflects a systematic gap rather than a prompt artifact, and (2) the outcome aligns with the prompt-robustness analysis in Section 5.1.

### D.6 COMPARISON OF NUMERIC SCORES ASSIGNED BY HUMAN AND AI REVIEWERS

While Section 5.5 focused on the misalignment between human and AI peer reviews from three commercial LLMs (GPT-4o, Gemini, and Claude) from the NeurIPS2022 review samples, this section presents the corresponding results for two open-source LLMs (Llama and Qwen), as shown in Figure S3. The main findings from GPT-4o, Gemini, and Claude also hold for these two open-source models, with one notable difference: Llama and Qwen exhibit an even larger divergence in Presentation scores than Claude, the most overly-positive one amongst the commercial LLMs across all categories. In terms of Contribution scores, the evaluations from Llama and Qwen were similar to those of Claude.

In addition, we examine data from three other conferences (NeurIPS2023, NeurIPS2024, and ICLR2024). Although the results from these conferences are slightly less reliable—given that human reviews may have been influenced by AI use following the release of ChatGPT—the overall trend persists: LLMs tend to inflate the quality of papers compared to human reviewers.

### D.7 VALIDATION OF AI-EDITED REVIEWS

To simulate varying degrees of LLM-assisted editing, we took human-written reviews and generated AI-edited versions at four levels of modification: *Minimum*, *Moderate*, *Extensive*, and *Maximum* (see Appendix E.3 for the editing prompts used). To validate that these prompts produce meaningfully distinct levels of change, we computed cosine similarity scores between each AI-edited review and its original human version using text embeddings. As shown in Table S14, similarity scores progressively decrease with greater editing intensity, confirming that the levels are semantically distinguishable.

Table S14: Similarity score of edited reviews and original human-written reviews using different prompts.

| Minimum | Moderate | Extensive | Maximum |
|---|---|---|---|
| 0.9841 | 0.9261 | 0.8616 | 0.6799 |

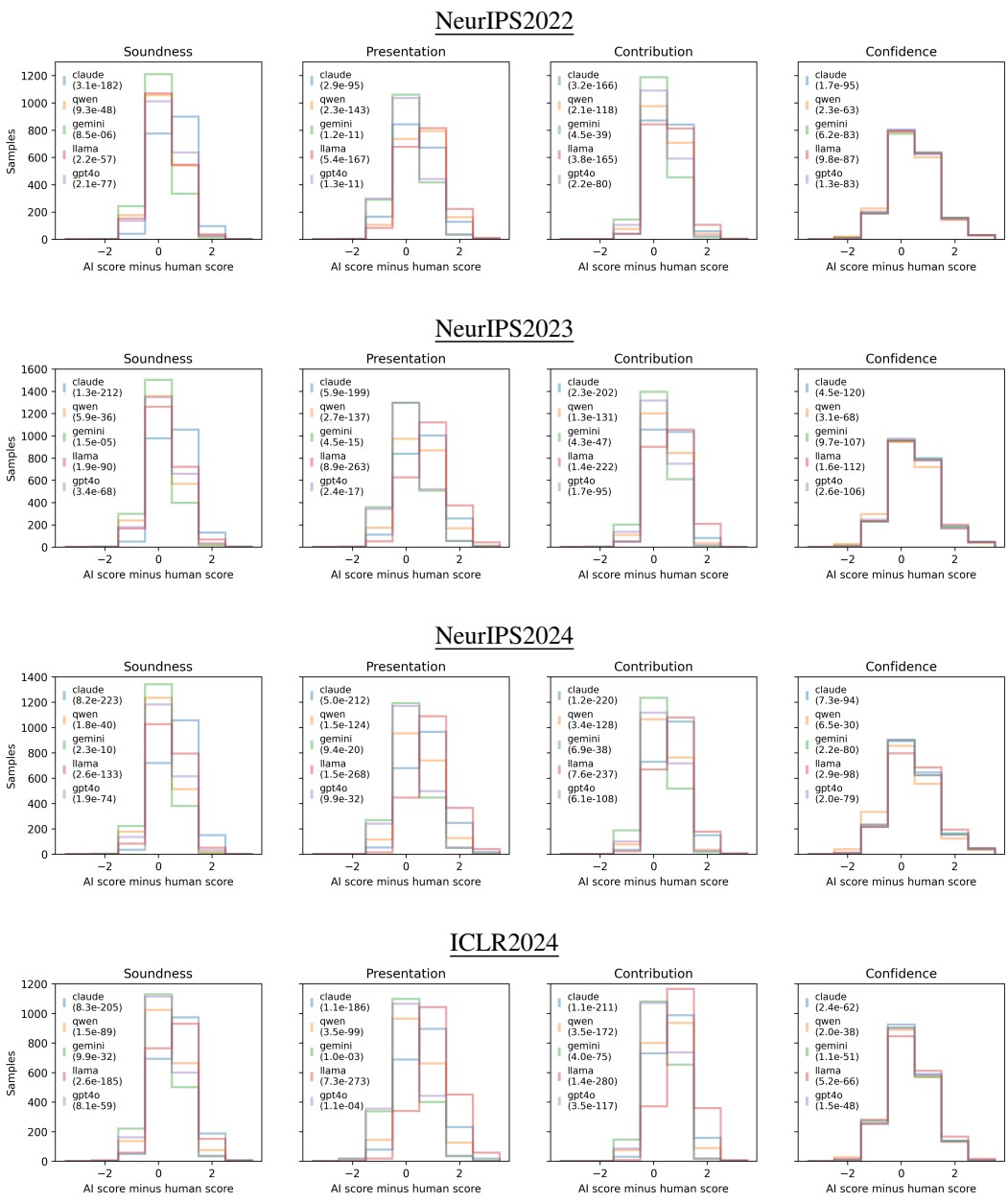

Figure S3: Difference between AI and human scores. For each matched review (aligned by paper ID and recommendation), score differences were computed and displayed as histograms. Scores range from 1 to 4 for all metrics except Confidence, which ranges from 1 to 5. Statistical significance was assessed using a two-sided Wilcoxon signed-rank test, with p-values shown in the legend. This figure includes only NeurIPS2022–2024 and ICLR2024, because they are the onyl conferences that required reviewers to submit these scores in their review templates.

## D.8   SENSITIVITY OF ANCHOR EMBEDDING DETECTABILITY PERFORMANCE TO EMBEDDING MODEL

In addition to the OpenAI embedding model used for our main Anchor embedding results, we conducted experiments using two other embeddings models: one based on a traditional encoder architecture (Jina AI embedding model v3 (Sturua et al., 2024)), and another based on a decoder architecture (Intfloat e5-mistral-7b-instruct (Wang et al., 2023; 2022)). We picked these based on their popularity (assessed via hugging face downloads) and their ability to accommodate long context lengths. The evaluation results for the Anchor embedding method when used with these models

Table S15: Anchor method detectability results using other embedding models

| Embedding Model | Test Reviews | Target FPR | Actual FPR | Actual TPR |
|---|---|---|---|---|
| Jina | GPT-4o | 0.1% | 0.2% | 50.5% |
| Jina | GPT-4o | 0.5% | 0.7% | 64.8% |
| Jina | GPT-4o | 1.0% | 1.1% | 71.4% |
| Jina | Gemini | 0.1% | 0.2% | 29.7% |
| Jina | Gemini | 0.5% | 0.7% | 46.2% |
| Jina | Gemini | 1.0% | 1.1% | 53.1% |
| Jina | Claude | 0.1% | 0.2% | 22.0% |
| Jina | Claude | 0.5% | 0.7% | 36.9% |
| Jina | Claude | 1.0% | 1.1% | 45.1% |
| e5-mistral | GPT-4o | 0.1% | 0.1% | 26.0% |
| e5-mistral | GPT-4o | 0.5% | 0.4% | 41.7% |
| e5-mistral | GPT-4o | 1.0% | 0.8% | 48.4% |
| e5-mistral | Gemini | 0.1% | 0.1% | 26.4% |
| e5-mistral | Gemini | 0.5% | 0.4% | 40.8% |
| e5-mistral | Gemini | 1.0% | 0.8% | 47.4% |
| e5-mistral | Claude | 0.1% | 0.1% | 26.6% |
| e5-mistral | Claude | 0.5% | 0.4% | 43.9% |
| e5-mistral | Claude | 1.0% | 0.8% | 51.9% |

on the test set are provided in Table S15. While we observe that the performance of the anchor embedding approach is lower using these alternative embedding models than when utilizing the OpenAI embedding model, its performance is still comparable to that of Binoculars. These results illustrate how the choice of the embedding model can impact detection performance of the Anchor approach.

### D.9    IMPACT OF FINETUNING SUPERVISED CLASSIFIERS ON AI PEER REVIEW DATASET

Our main results in Table 2 showed that existing detection methods which are based on finetuned classifiers perform poorly at detecting AI text in peer reviews. One reason for this could be due to a mismatch between the domain of text used to train these classifiers and our task of detecting AI text in the peer review domain. Therefore, we conducted experiments where we finetuned supervised classifiers on our dataset of paired human & AI-written peer reviews.

As the existing methods based supervised classifiers are all finetuned variants of RoBERTa, we similarly trained a RoBERTa classifier separately on peer reviews generated by GPT-4o, Claude, and Gemini to assess their out-of-domain detection accuracy. To maintain a fair comparison, we used the exact same calibration dataset (consisting of GPT-4o, Claude, and Gemini reviews) to calibrate the classification threshold as was used for our anchor embedding approach. The performance of the resulting classifiers is summarized in Table S16.

When tested on AI reviews generated by the same LLM as was used to construct the training dataset, we observe that these classifiers achieve near 100% TPR at an FPR < 1%. However, when evaluated on out-of-domain test data (i.e., peer reviews generated by LLMs other than the one used to construct the training dataset), the performance is highly variable and inconsistent. Thus, the performance of supervised methods is dependent upon the training dataset including samples from the same LLM which was used to generate the evaluated text.

Using the same calibration dataset as these supervised finetuning experiments, our Anchor embedding approach achieves overall strong performance across reviews generated by all LLMs without requiring any separate training. We also note that training supervised classifiers to achieve competitive performance in this task would not be possible without our paired human & AI-written peer review dataset, which is a core contribution of our work. Therefore, the improved performance of supervised methods when trained on our dataset (relative to their baseline implementations trained on general texts) can be viewed as further evidence of our dataset's value.

Table S16: Performance of RoBERTa classifiers trained & tested on different subsets of AI-generated peer reviews from our dataset

| Train Reviews | Test Reviews | Target FPR | Actual FPR | Actual TPR |
|---|---|---|---|---|
| GPT-4o | GPT-4o | 0.1% | 0.3% | 100% |
| GPT-4o | GPT-4o | 0.5% | 0.6% | 100% |
| GPT-4o | GPT-4o | 1.0% | 0.9% | 100% |
| GPT-4o | Gemini | 0.1% | 0.3% | 83.5% |
| GPT-4o | Gemini | 0.5% | 0.6% | 92.0% |
| GPT-4o | Gemini | 1.0% | 0.9% | 95.8% |
| GPT-4o | Claude | 0.1% | 0.3% | 44.9% |
| GPT-4o | Claude | 0.5% | 0.6% | 59.1% |
| GPT-4o | Claude | 1.0% | 0.9% | 70.4% |
| Gemini | GPT-4o | 0.1% | 0.4% | 99.9% |
| Gemini | GPT-4o | 0.5% | 0.8% | 99.6% |
| Gemini | GPT-4o | 1.0% | 1.4% | 99.9% |
| Gemini | Gemini | 0.1% | 0.4% | 100% |
| Gemini | Gemini | 0.5% | 0.8% | 100% |
| Gemini | Gemini | 1.0% | 1.4% | 100% |
| Gemini | Claude | 0.1% | 0.4% | 41.7% |
| Gemini | Claude | 0.5% | 0.8% | 49.8% |
| Gemini | Claude | 1.0% | 1.4% | 60.2% |
| Claude | GPT-4o | 0.1% | 0.5% | 95.9% |
| Claude | GPT-4o | 0.5% | 0.7% | 97.6% |
| Claude | GPT-4o | 1.0% | 0.9% | 98.3% |
| Claude | Gemini | 0.1% | 0.5% | 2.7% |
| Claude | Gemini | 0.5% | 0.7% | 3.0% |
| Claude | Gemini | 1.0% | 0.9% | 3.5% |
| Claude | Claude | 0.1% | 0.5% | 100% |
| Claude | Claude | 0.5% | 0.7% | 100% |
| Claude | Claude | 1.0% | 0.9% | 100% |

## D.10 Topic Distribution of Papers for Evaluated Reviews

We analyzed the keywords associated with 500 papers from our ICLR 2021 test set using metadata available for each paper in OpenReview. The statistic of the topics is given below.

1. Uncategorized: 19.5%

2. Deep Learning Foundations: 17.5%

3. Reinforcement Learning: 12.9%

4. Robustness & Adversarial M: 11.2%

5. Representation Learning: 8.0%

6. Optimization & Training: 8.0%

7. NLP & Multimodal: 6.0%

8. Computer Vision: 5.7%

9. Meta-Learning: 4.6%

10. Theory & Interpretability: 3.7%

11. Model Efficiency: 2.6%

This breakdown indicates that our benchmark is not dominated by any single domain, and therefore the observed differences between human-written and LLM-generated reviews cannot be attributed to a narrow topical bias in the underlying papers.

### D.11 How do common factors which are characteristic of AI peer reviews impact detectability?

Building upon our analysis of how human and AI-written peer reviews differ (Section 5.4 and Appendix D.4), we analyzed AI-written reviews from the ICLR 2021 test set for the presence of three common factors that we previously identified as characteristic of AI reviews: (1) a lack of references to prior work, (2) a lack of references to specific details in the paper, and (3) overly generic criticisms. We used an LLM-as-a-judge approach with GPT-4.1 to quantify which of these issues were present across the 500 analyzed reviews.

The most commonly identified factor among this set of reviews was the lack of references to prior work, which was present in over 99% of the cases where the review was flagged as AI-written by the Anchor method. The next most common factor was the lack of references to specific details in the paper, which was flagged in 26% of papers which were detected as AI-written. Finally, 22% of papers classified as AI-generated were determined to have overly generic criticisms. These results indicate that a lack of reference to prior work is the most prevalent factor present among AI-generated reviews.

Interestingly, we do not observe a correlation between the presence of these factors in an AI-generated review and the detection accuracy of the Anchor embedding method. This could be due to the already strong detection accuracy of the Anchor method which limits the number of false negatives (only 14.4% of AI-generated reviews were classified as human-written at a calibrated FPR of 0.1%). An alternative explanation could be that the Anchor embedding method relies on other, more nuanced linguistic differences than the factors which humans identify as being characteristic of AI-generated reviews (e.g., lack of references, generic criticisms).

## E Prompts

This section includes the prompts we used to generate AI peer review texts. Due to space limitations, we provide only the ICLR2022 review guideline and review template here. Those for other years and other conferences (e.g., NeurIPS) are available on the respective conference official websites[4].

### E.1 Prompts for Generating Reviews

System prompt:

```
You are an AI researcher reviewing a paper submitted to a prestigious AI research
conference.
You will be provided with the manuscript text, the conference's reviewer guidelines,
and the decision for the paper.
Your objective is to thoroughly evaluate the paper, adhering to the provided
guidelines, and return a detailed assessment that supports the given decision using
the specified response template.
Ensure your evaluation is objective, comprehensive, and aligned with the conference
standards.

{reviewer_guideline}

{review_template}
```

User prompt:

```
Here is the paper you are asked to review. Write a well-justified review of this
paper that aligns with a '{human_reviewer_decision}' decision.

```
{text}
```
```

---

[4]https://icml.cc/Conferences/{2016..2024}
and https://neurips.cc/Conferences/{2016..2024}

ICLR2022 Reviewer Guideline
({`reviewer_guideline`} in the system prompt):

## Reviewer Guidelines

1. Read the paper: It's important to carefully read through the entire paper, and to look up any related work and citations that will help you comprehensively evaluate it. Be sure to give yourself sufficient time for this step.

2. While reading, consider the following:
   - Objective of the work: What is the goal of the paper? Is it to better address a known application or problem, draw attention to a new application or problem, or to introduce and/or explain a new theoretical finding? A combination of these? Different objectives will require different considerations as to potential value and impact.
   - Strong points: is the submission clear, technically correct, experimentally rigorous, reproducible, does it present novel findings (e.g. theoretically, algorithmically, etc.)?
   - Weak points: is it weak in any of the aspects listed in b.?
   - Be mindful of potential biases and try to be open-minded about the value and interest a paper can hold for the entire ICLR community, even if it may not be very interesting for you.

3. Answer three key questions for yourself, to make a recommendation to Accept or Reject:
   - What is the specific question and/or problem tackled by the paper?
   - Is the approach well motivated, including being well-placed in the literature?
   - Does the paper support the claims? This includes determining if results, whether theoretical or empirical, are correct and if they are scientifically rigorous.

4. Write your initial review, organizing it as follows:
   - Summarize what the paper claims to contribute. Be positive and generous.
   - List strong and weak points of the paper. Be as comprehensive as possible.
   - Clearly state your recommendation (accept or reject) with one or two key reasons for this choice.
   - Provide supporting arguments for your recommendation.
   - Ask questions you would like answered by the authors to help you clarify your understanding of the paper and provide the additional evidence you need to be confident in your assessment.
   - Provide additional feedback with the aim to improve the paper. Make it clear that these points are here to help, and not necessarily part of your decision assessment.

5. General points to consider:
   - Be polite in your review. Ask yourself whether you'd be happy to receive a review like the one you wrote.
   - Be precise and concrete. For example, include references to back up any claims, especially claims about novelty and prior work
   - Provide constructive feedback.
   - It's also fine to explicitly state where you are uncertain and what you don't quite understand. The authors may be able to resolve this in their response.
   - Don't reject a paper just because you don't find it "interesting". This should not be a criterion at all for accepting/rejecting a paper. The research community is so big that somebody will find some value in the paper (maybe even a few years down the road), even if you don't see it right now.

ICLR2022 Review Template
({`reviewer_template`} in the system prompt):

## Response template (JSON format)

Provide the review in valid JSON format with the following fields. Ensure all fields are completed as described below. The response must be a valid JSON object.

- "summary_of_the_paper": Briefly summarize the paper and its contributions. This is not the place to critique the paper; the authors should generally agree with a well-written summary. You may use paragraphs and bulleted lists for formatting, but ensure that the content remains a single, continuous text block. Do not use nested JSON or include additional fields.

- "main_review": "Provide review comments as a single text field (a string). Consider including assessment on the following dimensions: a comprehensive list of strong and weak points of the paper, your recommendation, supporting arguments for your recommendation, questions to clarify your understanding of the paper or request additional evidence, and additional feedback with the aim to improve the paper. You may use paragraphs and bulleted lists for formatting, but ensure that the content remains a single, continuous text block. Do not use nested JSON or include additional fields."

- "summary_of_the_review": Concise summary of 'main_review'. You may use paragraphs and bulleted lists for formatting, but ensure that the content remains a single, continuous text block. Do not use nested JSON or include additional fields.

- "correctness": A numerical rating on the following scale to indicate that the claims and methods are correct. The value should be between 1 and 4, where:
    - 1 = The main claims of the paper are incorrect or not at all supported by theory or empirical results.
    - 2 = Several of the paper's claims are incorrect or not well-supported.
    - 3 = Some of the paper's claims have minor issues. A few statements are not well-supported, or require small changes to be made correct.
    - 4 = All of the claims and statements are well-supported and correct.

- "technical_novelty_and_significance": A numerical rating on the following scale to indicate technical novelty and significance. The value should be between 1 and 4, where:
    - 1 = The contributions are neither significant nor novel.
    - 2 = The contributions are only marginally significant or novel.
    - 3 = The contributions are significant and somewhat new. Aspects of the contributions exist in prior work.
    - 4 = The contributions are significant and do not exist in prior works.

- "empirical_novelty_and_significance": A numerical rating on the following scale to indicate empirical novelty and significance. The value should be between 1 and 4, or -999 if not applicable, where:
    - 1 = The contributions are neither significant nor novel.
    - 2 = The contributions are only marginally significant or novel.
    - 3 = The contributions are significant and somewhat new. Aspects of the contributions exist in prior work.
    - 4 = The contributions are significant and do not exist in prior works.
    - -999 = Not applicable.

- "flag_for_ethics_review": A boolean value (`true` or `false`) indicating whether there are ethical concerns in the work.

- "recommendation": A string indicating the final decision, which must strictly be one of the following options: 'strong reject', 'reject, not good enough', 'marginally below the acceptance threshold', 'marginally above the acceptance threshold', 'accept, good paper', or 'strong accept, should be highlighted at the conference'.

- "confidence": A nuemrical values to indicate how confident you are in your evaluation. The value should be between 1 and 5, where:
    - 1 = You are unable to assess this paper and have alerted the ACs to seek an opinion from different reviewers.
    - 2 = You are willing to defend your assessment, but it is quite likely that you did not understand the central parts of the submission or that you are unfamiliar with some pieces of related work. Math/other details were not carefully checked.

- 3 = You are fairly confident in your assessment. It is possible that you did
not understand some parts of the submission or that you are unfamiliar with some
pieces of related work. Math/other details were not carefully checked.
- 4 = You are confident in your assessment, but not absolutely certain. It is
unlikely, but not impossible, that you did not understand some parts of the
submission or that you are unfamiliar with some pieces of related work.
- 5 = You are absolutely certain about your assessment. You are very familiar
with the related work and checked the math/other details carefully.

### E.2 ANCHOR REVIEW GENERATION PROMPT

Systen prompt:

You are an AI research scientist tasked with reviewing paper submissions for a top
AI research conference. Carefully read the provided paper, then write a detailed
review following a common AI conference review format (e.g., including summary,
strengths and weakness, limitations, questions, suggestions for improvement). Make
sure to include recommendation for the paper, either 'Accept' or 'Reject'. Your
review should be fair and objective.

User prompt:

Here is the paper you are asked to review:
```
{text}
```

### E.3 EDITING PROMPTS

Minimal Editing:

Please proofread my review for typos and grammatical errors without altering the
content. Keep the original wordings as much as you can, except for typo or
grammatical  error.

Moderate Editing:

Please polish my review to improve sentence structure and readability while keeping
the original intent clear.

Extensive Editing:

Please rewrite my review into a polished, professional piece that effectively
communicates its main points.

Maximum Editing:

Please transform my review into a high quality piece, using professional language
and a polished tone. Please also extend my review with additional details from the
oringial paper.

### E.4 ARCHETYPE PROMPTS

"Balanced":

You provide fair, balanced, thorough, and constructive feedback, objectively
highlighting both the strengths and weaknesses of the paper. You maintain a high
standard for research in your decision-making process. However, even if your
decision is to reject, you offer helpful suggestions for improvement.}

"Conservative":

```
You generally prefer established methods and are skeptical of unproven (that is, new
or unconventional) approaches. While you maintain high standards and rigor, you are
critical of papers presenting new ideas without extensive evidence and thorough
validation against established baselines. You place significant emphasis on
methodological soundness and are cautious about endorsing innovations that haven't
been rigorously tested.}
```

"Innovative":

```
You highly value novelty and bold approaches, often prioritizing novel ideas over
methodological perfection. While you maintain high standards, you are willing to
overlook minor flaws or incomplete validations and may accept the paper, if the
paper introduces a significant new concept or direction. Conversely, you tend to be
less enthusiastic about papers that, despite thorough methodology and analysis,
offer only incremental improvements, and may recommend rejection for such
submissions.}
```

"Nitpicky":

```
You are a perfectionist who meticulously examines every aspect of the paper,
including minor methodological details, technical nuances, and formatting
inconsistencies. Even if a paper presents novel ideas or significant contributions,
you may still recommend rejection if you identify a substantial number of minor
flaws. Your stringent attention to detail can sometimes overshadow the broader
significance of the work in your decision-making process.}
```

## F    LIMITATIONS

Our dataset primarily focuses on two conferences, both within the computer science domain. To broaden its applicability and relevance, incorporating additional conferences from diverse research areas would be beneficial. While we designed our prompting strategy to encourage stylistic diversity, prompt choice still influences generation, and real-world use cases may involve a wider range of prompting styles than those tested. We conducted a prompt sensitivity analysis (Section 5.1) to evaluate the robustness of our dataset under prompt variation, though broader coverage remains an open direction. Our study also includes detection scenarios where LLMs revise or extend human-written reviews ("AI-edits-human"), but does not simulate the reverse case—where a human revises an AI-generated draft—due to the difficulty of sourcing domain experts to perform such edits at scale. We consider this an important direction for future work. Lastly, our main results are based on evaluations of three commercial LLMs. Given the rapid emergence of new models, conducting comprehensive experiments across all available LLMs is infeasible. In addition, we leverage an open-source platform to run baseline experiments, where performance may vary depending on the choice of surrogate models. However, given the large number of baselines we evaluate, performing an exhaustive search for the optimal surrogate model for each method would be prohibitively expensive. Therefore, we use the default settings.

While our anchor embedding method shows promising results, it is not without limitations. It is task-specific by design and is therefore unsuitable as a general-purpose AI text detection model. Additionally, its reliance on commercial LLM APIs may introduce challenges w.r.t. computational cost and scalability. The method's performance can also be sensitive to the choice of anchor LLMs. This is not specific to Anchor but reflects a general challenge in black-box detection settings where the source LLM is unknown. Acknowledging these limitations, our findings nonetheless highlight how leveraging manuscript context as auxiliary information can significantly improve the accuracy and robustness of AI-generated peer review detection, especially under low-FPR constraints.

## G    ARTIFACT USE CONSISTENT WITH INTENDED USE

In our work, we ensured that the external resources we utilized were applied in a manner that aligns with their intended purposes. We used several LLMs (including GPT-4o, Gemini, Claude, Qwen, and Llama) as well as an open-source package, IMGTB, with a focus on advancing research in a non-commercial, open-source context. The artifacts from our work will be non-commercial, for-research, and open-sourced.

## H    USE OF AI TOOL

GitHub Copilot and ChatGPT were used to aid in coding for analysis and in editing text for clarity.

