# OpenReview forum: "Is Your Paper Being Reviewed by an LLM? Benchmarking AI Text Detection in Peer Review"
_ICLR.cc/2026/Conference — ICLR 2026 Poster_

### Official Review · Reviewer_xwvH · 2025-10-29

**Soundness:** 2
**Presentation:** 2
**Contribution:** 2
**Rating:** 4
**Confidence:** 3

**Summary:**

This paper aims to detect peer reviews generated by LLMs. The authors construct a benchmark dataset, where there are both human reviews and AI-written reviews of papers submitted to computer conferences in recent years. This paper also proposes a detection method named "Anchor" to distinguish between such two types of peer reviews. Additionally, a series of evaluations are proposed to analyze the differences of the two types of review and the detection process, where the effectiveness of the proposed method is proved.

**Strengths:**

1. This paper targets at an ethical issues, detection of peer reviews from LLMs. Existing benchmarks mainly focus on general AI text detection, while this paper provides a new dataset for specific peer reviews, offering a useful benchmark for this problem.
2. In the evaluation part, the paper considers various aspects to analyze the problem, among which different prompts and LLM-assisted editing are meaningful. Furthermore, authors also discuss different types and parts of peer reviews, such as text-based reviews and score-based reviews, showing AI reviews' straight influence on paper acceptance.

**Weaknesses:**

1. There might be deviation in distribution of words because different papers have special domain biases. The paper lacks analysis of such diversity, which tends to amplify the differences between LLM-generated texts and human-written reviews. Whether usage of human-written reviews in a specific domain will improve the quality of AI-written reviews for similar papers has not been discussed here.
2. Despite "AI peer review detection robust to prompt variations" is described in section 5.2, four reviewer archetypes which lack diversity do not make sense enough. This paper does not analyze how people will write prompts in real situations via conducting user studies or analyzing over a series of real prompts.
3. The authors introduce several interesting aspects in the evaluation part, but the analysis merely remains at the level of describing certain phenomena. More detailed cases might be more convincing to show how much the phenomena or factors influence the detection to show your benchmark will be effective in peer review.

**Questions:**

1. Could the authors provide the diversity of the collected papers? Parts of the papers focus on specific application or domain and might be characteristic, and what is the proportion of such papers? Will the benchmark and proposed method be more effective if such diversity is considered?
2. Could the authors provide some detailed cases when there are changes to the analyzed factors? It is convincing if there are examples to show whether the changes of some factors might influence the detection and whether existing methods might fail to handle such changes. And the authors might analyze why the proposed method is better from concrete examples or whether some factors lead to some misclassification results.
3. Chapter 4 only has one subsection 4.1 (this subsection and its title might be useless). The authors introduce the proposed method in detail after the 'EXPERIMENTAL RESULTS' chapter and such results prove the method, where the writing logic is confusing. The authors might reorganize these parts.

---

> ### Author Response · Authors · 2025-11-27
>
> **(Part 1/3)**
>
> Thank you for recognizing how our work addresses an important ethical issue in the use of LLMs for peer review, the usefulness of our dataset as a benchmark, the variety of aspects that we analyze to address this problem, and the different parts and types of peer reviews which we address. We respond to questions and concerns raised in your review below. All additional content from our rebuttal responses will be incorporated into the final version of our manuscript.
>
> > There might be deviation in distribution of words because different papers have special domain biases. The paper lacks analysis of such diversity, which tends to amplify the differences between LLM-generated texts and human-written reviews…Could the authors provide the diversity of the collected papers? Parts of the papers focus on specific application or domain and might be characteristic, and what is the proportion of such papers? Will the benchmark and proposed method be more effective if such diversity is considered?
>
> Thank you for this suggestion. We analyzed the keywords associated with 500 papers from our ICLR 2021 test set using metadata available for each paper in OpenReview. The statistic of the topics is given below
>
> 1. Uncategorized:  19.5%
> 2. Deep Learning Foundations:17.5
> 3. Reinforcement Learning: 12.9%
> 4. Robustness & Adversarial M: 11.2%
> 5. Representation Learning: 8.0%
> 6. Optimization & Training: 8.0%
> 7. NLP & Multimodal: 6.0%
> 8. Computer Vision: 5.7%
> 9. Meta-Learning: 4.6%
> 10. Theory & Interpretability: 3.7%
> 11. Model Efficiency: 2.6%
>
> This breakdown indicates that our benchmark is not dominated by any single domain, and therefore the observed differences between human-written and LLM-generated reviews cannot be attributed to a narrow topical bias in the underlying papers.

---

> ### Author Response · Authors · 2025-11-27
>
> **Part (2/3)**
>
> > Despite "AI peer review detection robust to prompt variations" is described in section 5.2, four reviewer archetypes which lack diversity do not make sense enough. This paper does not analyze how people will write prompts in real situations via conducting user studies or analyzing over a series of real prompts.
>
> We appreciate your concern regarding the diversity of prompts used throughout our work. Regarding our prompt variations analysis described in section 5.2, the four reviewer archetypes represent only a single element of how we assessed the robustness of AI review detection to prompt variations.
>
> In lines 373-392 of section 5.2, we describe how we also evaluated the detectability of reviews generated by several recent agent-based or structured LLM review systems: AI Scientist (Lu et al., 2024), AgentReview (Jin et al., 2024), and DeepReview (Zhu et al., 2025). These systems incorporate features like multi-step workflows, reviewer personas, and self-reflection. Our analysis showed that despite the sophisticated varied prompting methods used by these approaches, Binoculars consistently achieve high AUROC (0.99), which is ∼0.03 larger than its performance on our own GPT-4o-generated baseline. This suggests that such advanced prompting or agentic strategies do not fundamentally degrade detectability, supporting the broader applicability of our benchmark.
>
> Beyond our analysis of review detection robustness to prompt evaluations, we also evaluated mixed-authorship generation settings with varying levels of AI editing of human-written peer reviews (Section 5.3). This alternative evaluation setting provides valuable insights into how AI text detection methods perform under different real-world use cases for LLMs in the peer review process.
>
> Additionally, during the rebuttal period, we conducted a new analysis of AI peer reviews generated from bullet points, which represents an alternative prompting scenario which could be used in practice. Specifically, we aimed to simulate the scenario of a human drafting bullet points and using an LLM to assist with writing the full review. We did this by first asking an LLM (GPT-4.1) to condense a human-written peer review into a series of bullet points which capture the core points of the review. We then prompted GPT-4.1 to generate a review solely from the resulting bullet points (which themselves were derived from human-written reviews sampled from the ICLR 2021 test set).
>
> The Anchor embedding method detection results for the 500 AI reviews generated from bullet points are provided in the table below. Consistent with our previous analysis of mixed-authorship generation settings (AI editing of human-written peer reviews), we find that these reviews which are generated from the core points of human-written peer reviews are much more challenging to detect. This generation setting could be viewed as a legitimate use case for LLMs in the peer review process because the core points and ideas of the review are still originating from a human, with the LLM serving as a writing assistant. Our main focus is the identification of unethical uses of LLMs in the peer review process (i.e., when it is used to fully generate the peer review); these results demonstrate how our Anchor method performs well at detecting such uses while having relatively low detection rates for cases where LLMs are primarily used for writing assistance.
>
> | Target FPR | Actual FPR | Actual TPR |
> | :------: | :------: | :------: |
> | 0.1% | 0.2% | 5.8% |
> | 0.5% | 0.8% | 14.4% |
> | 1.0% | 2.2% | 18.6% |
>
> The potential set of real-world prompts that could be used to produce AI peer reviews is vast and undefined; it is therefore impossible to construct a dataset with complete coverage of all possible prompting strategies. Nevertheless, our dataset spans a wide range of prompting scenarios which go beyond fully-synthetic reviews by covering other common use cases such as AI-editing and generation from provided bullet points.
>
> Finally, we agree that a user study would be valuable for characterizing the range of prompts which could be used in practice. However, recruiting participants for such a study presents challenges. Our main focus is on detecting unethical applications of LLMs in the peer review process. Therefore, the target population for participants in this study would be individuals who are willing to (1) admit that they engage in unethical behavior that violates the review policies of most major conferences, and (2) are willing to share the prompts that they use to engage in such behavior with researchers who are actively working to detect it. We greatly appreciate you raising this concern and will incorporate this discussion in the final version of our manuscript.

---

> ### Author Response · Authors · 2025-11-27
>
> **(Part 3/3)**
> > The authors introduce several interesting aspects in the evaluation part, but the analysis merely remains at the level of describing certain phenomena. More detailed cases might be more convincing to show how much the phenomena or factors influence the detection to show your benchmark will be effective in peer review.
> >
> >*(and)*
> >
> > Could the authors provide some detailed cases when there are changes to the analyzed factors? It is convincing if there are examples to show whether the changes of some factors might influence the detection and whether existing methods might fail to handle such changes. And the authors might analyze why the proposed method is better from concrete examples or whether some factors lead to some misclassification results.
>
> Thank you for this suggestion. Building upon our analysis of how human and AI-written peer reviews differ (Section 5.4 and Appendix D.4), we analyzed AI-written reviews from the ICLR 2021 test set for the presence of three common factors that we previously identified as characteristic of AI reviews: (1) a lack of references to prior work, (2) a lack of references to specific details in the paper, and (3) overly generic criticisms. We used an LLM-as-a-judge approach with GPT-4.1 to quantify which of these issues were present across the 500 analyzed reviews.
>
> The most commonly identified factor among this set of reviews was the lack of references to prior work, which was present in over 99% of the cases where the review was flagged as AI-written by the Anchor method. The next most common factor was the lack of references to specific details in the paper, which was flagged in 26% of papers which were detected as AI-written. Finally, 22% of papers classified as AI-generated were determined to have overly generic criticisms. These results indicate that a lack of reference to prior work is the most prevalent factor present among AI-generated reviews.
>
> Interestingly, we do not observe a correlation between the presence of these factors in an AI-generated review and the detection accuracy of the Anchor embedding method. This could be due to the already strong detection accuracy of the Anchor method which limits the number of false negatives (only 14.4% of AI-generated reviews were classified as human-written at a calibrated FPR of 0.1%). An alternative explanation could be that the Anchor embedding method relies on other, more nuanced linguistic differences than the factors which humans identify as being characteristic of AI-generated reviews (e.g., lack of references, generic criticisms). We appreciate your suggestion for this analysis and will include it in the final version of our paper.
>
> > Chapter 4 only has one subsection 4.1 (this subsection and its title might be useless). The authors introduce the proposed method in detail after the 'EXPERIMENTAL RESULTS' chapter and such results prove the method, where the writing logic is confusing. The authors might reorganize these parts.
>
> Thank you for pointing out this issue with the structure of Chapter 4. We have addressed this in our revised manuscript by reorganizing Section 4. Specifically, we moved the description of the Anchor method (previously Section 5.1, now Section 4.1) to appear before the discussion of the main experimental results (Section 4.2)

---

### Official Review · Reviewer_6PDK · 2025-11-01

**Soundness:** 3
**Presentation:** 3
**Contribution:** 3
**Rating:** 6
**Confidence:** 4

**Summary:**

This paper addresses the growing concern that LLMs are being used to generate peer reviews for academic conferences. The authors created a large-scale dataset for AI-generated peer reviews and benchmarked 18 existing AI text detection methods to assess their effectiveness at identifying LLM-written reviews, showing that most existing detection methods struggle significantly at low false positive rates. The proposed "Anchor" detection method uses manuscript context by comparing test reviews to reference AI reviews generated for the same paper, which is simple but effective. The authors also identified several differences between human and AI-written reviews, and conducted robustness analyses to demonstrate that detection performance remains stable across different prompting strategies.

**Strengths:**

1. The paper addresses a critical and emerging problem in scientific peer review, the detectability of LLM-generated reviews.

2. The constructed dataset is large-scale and can be useful for the ongoing studies of AI text detection. The authors benchmarks 18 existing AI text detection methods, showing that most existing detection methods struggle significantly at low false positive rates.

3. I like how authors study the differences between human and AI-written reviews. This could be very helpful for understanding the limitations of LLM-generated peer reviews and developing better detection methods.

4. The overall structure is well organized and the contents are sufficient to show the challenges in detecting LLM-generated reviews.

**Weaknesses:**

1. One potential explanation for the poor performance is that peer review text may be out-of-distribution for these detection models, have the authors considered evaluating whether fine-tuning existing detection methods on peer review data could substantially improve detectability?

2. Although the authors tested the robustness of their methods across different prompts and pipelines, they all follow the same paradigm: given a paper, generate reviews on it. However, in reality, this may not be how reviewers actually use LLMs to generate reviews. For example, a reviewer might provide some bullet points to an LLM and ask it to expand on them [1].

3. A potential weakness of the Anchor method is its reliance on a reference dataset of AI-generated reviews, which may quickly lose its effectiveness as AI evolves.

[1] Lee, Mina, et al. "A design space for intelligent and interactive writing assistants." Proceedings of the 2024 CHI Conference on Human Factors in Computing Systems. 2024.

**Questions:**

See Weakness.

---

> ### Author Response · Authors · 2025-11-27
>
> **(Part 1/2)**
>
> Thank you for recognizing the importance of the problem we address, the scale and usefulness of our dataset, our study of the differences between human and AI-written peer reviews, and the overall structure & content of our paper. We respond to questions and concerns raised in your review below. All additional content from our rebuttal responses will be incorporated into the final version of our manuscript.
>
> > One potential explanation for the poor performance is that peer review text may be out-of-distribution for these detection models, have the authors considered evaluating whether fine-tuning existing detection methods on peer review data could substantially improve detectability?
>
> We appreciate this suggestion and have conducted fine-tuning experiments for existing supervised detection methods during the rebuttal period. As the existing methods are all finetuned variants of RoBERTa, we similarly trained a RoBERTa classifier separately on peer reviews generated by GPT-4o, Claude, and Gemini to assess their out-of-domain detection accuracy. To maintain a fair comparison, we used the exact same calibration dataset (consisting of GPT-4o, Claude, and Gemini reviews) to calibrate the classification threshold as was used for our anchor embedding approach.
>
> When tested on AI reviews generated by the same LLM as was used to construct the training dataset, we observe that these classifiers achieve near 100% TPR at an FPR < 1%. However, when evaluated on out-of-domain test data (i.e., peer reviews generated by LLMs other than the one used to construct the training dataset), the performance is highly variable and inconsistent. Thus, the performance of supervised methods is dependent upon the training dataset including samples from the same LLM which was used to generate the evaluated text. We provide complete results for the performance of the classifier trained and evaluated on reviews generated from different LLMs in the three tables below.
>
> | Train Reviews | Test Reviews | Target FPR | Actual FPR | Actual TPR |
> | :------: | :------: | :------: | :------: | :------: |
> | GPT-4o | GPT-4o | 0.1% | 0.3% | 100% |
> | GPT-4o | GPT-4o | 0.5% | 0.6% | 100% |
> | GPT-4o | GPT-4o | 1.0% | 0.9% | 100% |
> | GPT-4o | Gemini | 0.1% | 0.3% | 83.5% |
> | GPT-4o | Gemini | 0.5% | 0.6% | 92.0% |
> | GPT-4o | Gemini | 1.0% | 0.9% | 95.8% |
> | GPT-4o | Claude | 0.1% | 0.3% | 44.9% |
> | GPT-4o | Claude | 0.5% | 0.6% | 59.1% |
> | GPT-4o | Claude | 1.0% | 0.9% | 70.4% |
>
> | Train Reviews | Test Reviews | Target FPR | Actual FPR | Actual TPR |
> | :------: | :------: | :------: | :------: | :------: |
> | Gemini | GPT-4o | 0.1% | 0.4% | 99.9% |
> | Gemini | GPT-4o | 0.5% | 0.8% | 99.6% |
> | Gemini | GPT-4o | 1.0% | 1.4% | 99.9% |
> | Gemini | Gemini | 0.1% | 0.4% | 100% |
> | Gemini | Gemini | 0.5% | 0.8% | 100% |
> | Gemini | Gemini | 1.0% | 1.4% | 100% |
> | Gemini | Claude | 0.1% | 0.4% | 41.7% |
> | Gemini | Claude | 0.5% | 0.8% | 49.8% |
> | Gemini | Claude | 1.0% | 1.4% | 60.2% |
>
> | Train Reviews | Test Reviews | Target FPR | Actual FPR | Actual TPR |
> | :------: | :------: | :------: | :------: | :------: |
> | Claude | GPT-4o | 0.1% | 0.5% | 95.9% |
> | Claude | GPT-4o | 0.5% | 0.7% | 97.6% |
> | Claude | GPT-4o | 1.0% | 0.9% | 98.3% |
> | Claude | Gemini | 0.1% | 0.5% | 2.7% |
> | Claude | Gemini | 0.5% | 0.7% | 3.0% |
> | Claude | Gemini | 1.0% | 0.9% | 3.5% |
> | Claude | Claude | 0.1% | 0.5% | 100% |
> | Claude | Claude | 0.5% | 0.7% | 100% |
> | Claude | Claude | 1.0% | 0.9% | 100% |
>
> Using the same calibration dataset as these supervined finetuning experiments, our Anchor embedding approach achieves overall strong performance across reviews generated by all LLMs without requiring any separate training. We also note that training supervised classifiers to achieve competitive performance in this task would not be possible without our paired human & AI-written peer review dataset, which is a core contribution of our work. Therefore, the improved performance of supervised methods when trained on our dataset (relative to their baseline implementations trained on general texts) can be viewed as further evidence of our dataset’s value.

---

> ### Author Response · Authors · 2025-11-27
>
> **(Part 2/2)**
>
> > Although the authors tested the robustness of their methods across different prompts and pipelines, they all follow the same paradigm: given a paper, generate reviews on it. However, in reality, this may not be how reviewers actually use LLMs to generate reviews. For example, a reviewer might provide some bullet points to an LLM and ask it to expand on them [1].
>
> Thank you for this suggestion. In Table 2, we analyze a different paradigm than the “given a paper, generate reviews on it” setting used for our main results. Specifically, we analyze the impact of various levels of “AI editing” of human-written peer reviews. Our results show that as the amount of editing increases, so does the likelihood that the review will be flagged as AI-written. This alternative evaluation setting provides valuable insights into how AI text detection methods perform under different common use cases.
>
> We appreciate your suggestion to investigate the generation of AI reviews from bullet points provided to an LLM, and will include this additional reference in the final version of our manuscript. While we determined that it would be too time consuming to recruit human participants for this study during the rebuttal period, we conducted an experiment to simulate this generation setting by first asking an LLM (GPT-4.1) to condense a human-written peer review into a series of bullet points which capture the core points of the review. We then prompted an LLM to generate a review solely from the resulting bullet points (which themselves were derived from human-written reviews sampled from the ICLR 2021 test set).
>
> The Anchor embedding method detection results for the 500 AI reviews generated from bullet points are provided in the table below. Consistent with our previous analysis of mixed-authorship generation settings (AI editing of human-written peer reviews), we find that these reviews which are generated from the core points of human-written peer reviews are much more challenging to detect. This generation setting could be viewed as a legitimate use case for LLMs in the peer review process because the core points and ideas of the review are still originating from a human, with the LLM serving as a writing assistant. Our main focus is the identification of unethical uses of LLMs in the peer review process (i.e., when it is used to fully generate the peer review); these results demonstrate how our Anchor method performs well at detecting such uses while having relatively low detection rates for cases where LLMs are primarily used for writing assistance.
>
> | Target FPR | Actual FPR | Actual TPR |
> | :------: | :------: | :------: |
> | 0.1% | 0.2% | 5.8% |
> | 0.5% | 0.8% | 14.4% |
> | 1.0% | 2.2% | 18.6% |
>
> > A potential weakness of the Anchor method is its reliance on a reference dataset of AI-generated reviews, which may quickly lose its effectiveness as AI evolves.
>
> While our Anchor method can indeed be used with a reference dataset, our implementation allows for the anchor reviews to be dynamically generated at evaluation time by any LLM (see L317 of Section 5.1). Therefore, our method can remain relevant as AI evolves by using the latest frontier models to generate the anchor reviews rather than simply relying on a static reference dataset.

---

### Official Review · Reviewer_9j4M · 2025-11-05

**Soundness:** 3
**Presentation:** 3
**Contribution:** 2
**Rating:** 8
**Confidence:** 3

**Summary:**

This paper studies the problem of detecting AI-generated text in the peer-review process.

Its contribution includes:

1. A comprehensive dataset containing a total of 788,984 AI-written peer reviews paired with corresponding human reviews for ICLR and NeurIPS papers (2016–2024)
2. Benchmarks 18 existing AI-text detection methods 3 Proposes a simple context-aware “Anchor” approach that leverages manuscript content by comparing the semantic similarity between a target review and an LLM-generated reference review.
3. Share some interesting insight in stylistic differences between human and AI reviews.

**Strengths:**

1. The paper shows a certain level of originality. It creates a large-scale dataset in a new domain - peer review. Existing detection datasets e.g., HC3, RAID-TD, M4, Beemo, GRiD does not cover this domain. Also, the benchmarking framework’s focus on achieving reliable detection under low false-positive constraints is also a meaningful and novel contribution, as few prior works explicitly address this important practical and ethical consideration in review integrity contexts.

2. The paper is clearly written and easy to follow. The experimental setup is rigorous, with careful exclusion of post-2022 reviews to minimize potential LLM contamination. It provides extensive benchmarking across 18 detection methods. It also includes thoughtful analyses on prompt robustness, AI-assisted editing, and behavioral differences between human and AI reviewers.

**Weaknesses:**

1. The proposed Anchor method offers only incremental innovation. Its core idea—measuring embedding similarity between a suspected review and an LLM-generated reference—is conceptually similar to prior work such as DetectLLM and DNA-GPT, which also compare generated and reference texts to assess authorship likelihood.

2. The dataset also relies entirely on synthetic AI reviews, which may not fully represent real-world LLM-assisted peer-review behavior. Also, if the same or related model family (e.g., GPT-4) is used for both, the high similarity might simply reflect shared model characteristics rather than meaningful evidence of AI authorship.

**Questions:**

1. Did you test alternative embedding models or similarity metrics (e.g., sentence-transformers or language-agnostic encoders) to ensure the Anchor method’s robustness across representations?

2. Have you explored the “human revises AI draft” scenario, where an LLM-generated review is partially edited or rewritten by a human? How does the method perform under this mixed authorship condition?

**Details Of Ethics Concerns:**

No ethics concerns listed above

---

> ### Author Response · Authors · 2025-11-27
>
> **(Part 1/3)**
>
> Thank you for recognizing the comprehensiveness of our dataset, our benchmarking of 18 existing AI text detection methods, the interesting insights we provide regarding stylistic differences between human and AI-written reviews, our paper’s originality, and its clarity. We respond to questions and concerns raised in your review below. All additional content from our rebuttal responses will be incorporated into the final version of our manuscript.
>
> > The proposed Anchor method offers only incremental innovation. Its core idea—measuring embedding similarity between a suspected review and an LLM-generated reference—is conceptually similar to prior work such as DetectLLM and DNA-GPT, which also compare generated and reference texts to assess authorship likelihood.
>
> Thank you for raising this concern. While we acknowledge a level of conceptual similarity between our Anchor method and these prior works, there are key differences in both the functionality and applicability of our approach relative to DetectLLM and DNA-GPT.
>
> DetectLLM assumes a white-box detection setting where the detector has access to the LLM used to generate the evaluated text. Unfortunately this setting is unrealistic for our task of detecting AI text in peer review because there is no way of knowing which LLM was used to generate the review. Additionally, we view frontier commercial models as the most likely to be used in practice due to their superior capabilities; since these models are only available via an API, methods such as DetectLLM which require full access to the model are not compatible.
>
> Another key difference between our Anchor method and both DetectLLM and DNA-GPT is that it leverages additional context which is available in the peer review setting: the manuscript of the paper which is being reviewed. In contrast, DNA-GPT truncates the middle of an evaluated text, regenerates the remainder with an LLM, and then analyzes the difference between the original & regenerated portions via n-gram analysis (in the black-box detection setting). This truncation & regeneration setting is unlikely to produce text which is as representative of AI-generated reviews because it lacks grounding in the source content (the paper). To the best of our knowledge, our anchor embedding approach is unique among AI text detection methods in its use of the additional context available in the peer review setting.
>
> We appreciate you raising this point and will include this additional discussion in Section 2 of our revised manuscript.

---

> ### Author Response · Authors · 2025-11-27
>
> **(Part 2/3)**
>
> > The dataset also relies entirely on synthetic AI reviews, which may not fully represent real-world LLM-assisted peer-review behavior. Also, if the same or related model family (e.g., GPT-4) is used for both, the high similarity might simply reflect shared model characteristics rather than meaningful evidence of AI authorship.
>
> We evaluated multiple different prompting strategies throughout our work, ranging from prompts which produce fully AI-written reviews to ones which request varying levels of LLM editing of human-written peer reviews (Section 5.3). Our primary focus is on the former scenario of detecting fully AI-written peer review, which we believe to be the most pressing ethical problem. However, we also included the LLM-edited human reviews (with varying levels of editing) from our analysis in Section 5.3 in our released dataset.
>
> Additionally, during the rebuttal period, we conducted a new analysis of AI peer reviews generated from bullet points, which represents an alternative prompting scenario which could be used in practice. Specifically, we aimed to simulate the scenario of a human drafting bullet points and using an LLM to assist with writing the full review. We did this by first asking an LLM (GPT-4.1) to condense a human-written peer review into a series of bullet points which capture the core points of the review. We then prompted GPT-4.1 to generate a review solely from the resulting bullet points (which themselves were derived from human-written reviews sampled from the ICLR 2021 test set).
>
> The Anchor embedding method detection results for the 500 AI reviews generated from bullet points are provided in the table below. Consistent with our previous analysis of mixed-authorship generation settings (AI editing of human-written peer reviews), we find that these reviews which are generated from the core points of human-written peer reviews are much more challenging to detect. This generation setting could be viewed as a legitimate use case for LLMs in the peer review process because the core points and ideas of the review are still originating from a human, with the LLM serving as a writing assistant. Our main focus is the identification of unethical uses of LLMs in the peer review process (i.e., when it is used to fully generate the peer review); these results demonstrate how our Anchor method performs well at detecting such uses while having relatively low detection rates for cases where LLMs are primarily used for writing assistance.
>
> | Target FPR | Actual FPR | Actual TPR |
> | :------: | :------: | :------: |
> | 0.1% | 0.2% | 5.8% |
> | 0.5% | 0.8% | 14.4% |
> | 1.0% | 2.2% | 18.6% |
>
>
> The potential set of real-world prompts that could be used to produce AI peer reviews is vast and undefined; it is therefore impossible to construct a dataset with complete coverage of all possible prompting strategies. Nevertheless, our dataset spans a wide range of prompting scenarios which go beyond fully-synthetic reviews by covering other common use cases such as AI-editing and generation from provided bullet points.

---

> ### Author Response · Authors · 2025-11-27
>
> **(Part 3/3)**
>
> > Did you test alternative embedding models or similarity metrics (e.g., sentence-transformers or language-agnostic encoders) to ensure the Anchor method’s robustness across representations?
>
> Thank you for this suggestion. During the rebuttal period, we conducted experiments using two other embeddings models: one based on a traditional encoder architecture (Jina AI embedding model v3), and another based on a decoder architecture (Intfloat e5-mistral-7b-instruct). We picked these based on their popularity (assessed via hugging face downloads) and their ability to accommodate long context lengths. The evaluation results for the Anchor embedding method when used with these models on the ICLR 2021 test set are provided in the tables below. While we observe that the performance of the anchor embedding approach is lower using these alternative embedding models than when utilizing the OpenAI embedding model, its performance is still comparable to that of Binoculars. We will include these additional results in our final manuscript along with a discussion of how the choice of the embedding model can impact detection performance.
>
> | Embed Model | Test Reviews | Target FPR | Actual FPR | Actual TPR |
> | :------ | :------: | :------: | :------: | :------: |
> | Jina | GPT-4o | 0.1% | 0.2% | 50.5% |
> | Jina | GPT-4o | 0.5% | 0.7% | 64.8% |
> | Jina | GPT-4o | 1.0% | 1.1% | 71.4% |
> | Jina | Gemini | 0.1% | 0.2% | 29.7% |
> | Jina | Gemini | 0.5% | 0.7% | 46.2% |
> | Jina | Gemini | 1.0% | 1.1% | 53.1% |
> | Jina | Claude | 0.1% | 0.2% | 22.0% |
> | Jina | Claude | 0.5% | 0.7% | 36.9% |
> | Jina | Claude | 1.0% | 1.1% | 45.1% |
>
> | Embed Model | Test Reviews | Target FPR | Actual FPR | Actual TPR |
> | :------ | :------: | :------: | :------: | :------: |
> | e5-mistral | GPT-4o | 0.1% | 0.1% | 26.0% |
> | e5-mistral | GPT-4o | 0.5% | 0.4% | 41.7% |
> | e5-mistral | GPT-4o | 1.0% | 0.8% | 48.4% |
> | e5-mistral | Gemini | 0.1% | 0.1% | 26.4% |
> | e5-mistral | Gemini | 0.5% | 0.4% | 40.8% |
> | e5-mistral | Gemini | 1.0% | 0.8% | 47.4% |
> | e5-mistral | Claude | 0.1% | 0.1% | 26.6% |
> | e5-mistral | Claude | 0.5% | 0.4% | 43.9% |
> | e5-mistral | Claude | 1.0% | 0.8% | 51.9% |
>
> > Have you explored the “human revises AI draft” scenario, where an LLM-generated review is partially edited or rewritten by a human? How does the method perform under this mixed authorship condition?
>
> While we have not explicitly analyzed a “human revises AI draft” scenario, we did investigate mixed authorship settings via our “AI edits human draft” experiments in Table 2. We found that the more drastic the level of AI editing, the greater the likelihood that our Anchor method will flag the review as AI-generated. We expect to see an inverse relationship in the “human revises AI draft” setting, where greater levels of human editing will reduce the likelihood of the review being flagged as AI-written. Due to the human labor requirements of conducting such a study, we were unable to conduct this analysis as part of our work. However, we will include a discussion of this opportunity for future studies on mixed human-AI authorship of peer reviews in the final version of our manuscript.

---

### Author Response · Authors · 2025-12-02
**Overview of changes in our revised manuscript**

Dear reviewers, ACs, SACs, and PCs,

We greatly appreciate the valuable feedback provided by the reviewers as well as their time and efforts to critique our work. We have provided responses to each review in the comment threads below. Additionally, we have uploaded a revised version of our manuscript which incorporates all new content and clarifications from our rebuttal. These changes are highlighted in blue in the revised manuscript; below we provide a summary of the key changes in the revised PDF:

- Section 2: Discussion of how our Anchor method differs from DetectLLM and DNA-GPT (**Reviewer 9j4M, W1**)
- Section 3.2: Clarification that our released dataset contains LLM-edited peer reviews, in addition to fully synthetic AI generated reviews (**Reviewer 9j4M, W2**)
- Section 4: Reorganized by moving description of the Anchor method (previously Section 5.1, now Section 4.1) to appear before the discussion of the main experimental results (Section 4.2) (**Reviewer xwvH, Q3**)
- Section 5.2: Discussion of the opportunity for future work investigating a “human revises AI draft” scenario (**Reviewer 9j4M, Q2**)
- Section 5.3: Analysis of the detectability of mixed-authorship peer reviews where an LLM generates the review from a set of bullet points (**Reviewer 6PDK, W2; Reviewer xwvH, W2; Reviewer 9j4M, W2**)
- Ethics Statement: Discussion of the ethical challenges associated with conducting user studies on how humans prompt LLMs to produce AI-generated reviews (**Reviewer xwvH, W2**)
- Appendix D.8: Analysis on the sensitivity of the Anchor method to the choice of the embedding model (**Reviewer 9j4M, Q1**)
- Appendix D.9: Analysis on the impact of finetuning supervised classifiers on our AI peer review dataset (**Reviewer 6PDK, W1**)
- Appendix D.10: Analysis on the distribution of topics for papers utilized in our work (**Reviewer xwvH, W1 / Q1**)
- Appendix D.11: Analysis on how common factors which are characteristic of AI peer reviews impact detectability (**Reviewer xwvH, W3 / Q2**)

We believe these substantial revisions have significantly strengthened our manuscript and thank the reviewers for all of their contributions.

Sincerely,

Paper 20658 Authors

---

### Public Comment · ~Sandeep_Kumar9 · 2026-02-25
**Related Work**

Dear author, Congratulations on your acceptance, We found the topic timely and important.

We would like to note that the regeneration-based detection method described in Section 4.1 — where an LLM-generated “anchor review” is compared to a candidate review using embeddings and cosine similarity thresholding — is conceptually and methodologically related to the regeneration-based framework introduced in our EMNLP 2024 paper, “‘Quis custodiet ipsos custodes?’ Who Will Watch the Watchmen? On Detecting AI-Generated Peer Reviews”  (https://aclanthology.org/2024.emnlp-main.1262/).

 In both works, a review is regenerated using an LLM under a similar prompt, embeddings are computed for the regenerated and candidate reviews, cosine similarity is measured between them, and a threshold-based decision rule is applied, based on the shared hypothesis that AI-generated reviews exhibit semantic consistency under regeneration.

As our EMNLP 2024 paper predates this submission and introduces this detection paradigm in the peer review setting, citing it would help situate the contribution within the existing literature and provide readers with a more complete view of prior work in this area. As the paper is on arXiv, we previously reached out but did not receive a response, so we are raising this here for completeness of the scholarly record.

---

> ### Public Comment · ~Sungduk_Yu1 · 2026-03-01
>
> Thank you for the thoughtful comment and for pointing out your related work. We agree this is an important and timely direction. We have added a citation to your EMNLP 2024 paper in the camera-ready version to help situate the broader line of work.

---

### Meta-Review · Area_Chair_wkC5 · 2025-12-22

**Summary:**

Reviewers question the limited novelty of Anchor, noting its similarity to prior reference-comparison approaches. The synthetic nature of AI reviews raises concerns about ecological validity and potential distributional artifacts. Missing baselines such as fine-tuned detectors or in-context LLM detectors leave open whether simpler approaches could match or exceed performance. Anchor’s dependence on a static reference dataset may weaken over time as models evolve. Some analyses lack depth, and prompt design does not reflect the diversity of real reviewer workflows. Overall, the dataset contribution is strong, but the detection method is incremental and generalizability remains uncertain.

**Reviewer Concerns:**

Most of the concerns have been well addressed.

**Reviewer Scores:**

8,6,4->6

---

### Decision · Program_Chairs · 2026-01-26

Accept (Poster)